# Similarity Group Equivariant Convolutional Networks

## Abstract

We introduce similarity group equivariant convolutional networks (SECNNs), designed to achieve continuous translation, rotation and scale equivariance, or discrete similarity group equivariance that involves discrete Dihedral group. The networks are implemented as steerable CNNs by employing a steerable and approximately shiftable and scalable basis for continuous translating, rotating and scaling convolution kernels within a five-dimensional position-orientation-scale-reflection space. Our results demonstrate that SECNNs attain state-of-the-art results on translated, rotated and scaled MNIST datasets. SECNNs also achieve the accuracy of other leading group equivariant networks on CIFAR10/100, while being equivariant to the full range of the similarity group in comparison to existing state of the art, which is equivariant to only sub-groups of the similarity group.

## 1 Introduction

Convolutional Neural Networks (CNNs) have played a significant role in the progress of deep learning, notably in tasks involving image and pattern recognition. The central operation in CNNs, convolution, effectively reduces the number of free parameters, allowing the network to be deeper with fewer parameters. Furthermore, convolution exhibits a property known as *equivariance*—a translation in the input results in a corresponding translation in the output. This attribute allows CNNs to recognize objects in various positions without requiring training on images with the object at different locations, thus, when combined with pooling, CNNs achieve translation-invariant recognition.

While translation equivariance is beneficial, real-world data often contains more complex transformations. Recognizing this, Cohen & Welling (2016) expanded the concept of equivariance in CNNs to include other transformations such as rotation and reflection, collectively known as the Euclidean group. This led to the development of group equivariant CNNs (G-CNNs) that employ group convolutions (G-convolutions) to handle transformations within this group. Crucially, this expansion into rotation introduced an extra orientation-specific dimension, resulting in both the feature maps and convolutional kernels of G-CNNs being inherently three-dimensional (3D).

Differing from traditional CNNs, G-CNNs must specifically account for transformations beyond translation. Special consideration must be taken because image pixels are typically sampled on a rectangular grid, whereas discrete translation equates to a mere shift in indices, transformations like rotation and scaling involve more complex interpolations. To facilitate continuous rotation, Worrall et al. (2017) employed circular harmonics as basis functions for representing convolution kernels. This basis approach treats a convolution kernel as a linear combination of these basis functions. Consequently, the convolution operation between an input and a kernel transforms into a two-step process: firstly, convolving the input with the basis functions, followed by a linear combination of the resultant feature maps. This methodology, a.k.a. steerable CNNs Cohen & Welling (2017); Weiler & Cesa (2019), allows the filter to undergo continuous rotation by simply manipulating the basis functions, which involves their multiplication with complex numbers.

Continuous rotation can be readily achieved since orientation is periodic, allowing the use of circular harmonics as a basis. However, achieving continuous equivariance in translation and scale presents greater challenges in G-CNNs, as functions of position and scale are typically aperiodic, leading mainly to discrete equivariance. Azulay & Weiss (2019) highlighted a limitation in the translation equivariance of conventional CNNs due to the discrete convolution and the process of downsampling. Recently, Sun & Blu (2023) introduced an innovative approach using a basis of harmonics defined

within a log-polar coordinate system, aiming for continuous scale equivariance. Nevertheless, fully realizing continuous equivariance across translation, rotation, and scale in G-CNNs is an ongoing challenge in the field. The inclusion of reflection, alongside these three transformations, constitutes what is known as the similarity transformation group. Importantly, similarity transformations do not alter the shape of an input. Thus, a convolutional network that is equivariant to the similarity transformation group can effectively preserve shape information, offering a promising direction for research involving similarity transformations, *e.g.,* feature matching Gleize et al. (2023).

In addition to being able to handle combined continuous similarity transformations, the difficulty of designing similarity group equivariant convolutional networks is the computational cost associated with multi-dimensional kernels and feature maps. G-convolutions about rotation, scaling and reflection introduce new dimensions to account for orientation, scale and reflection respectively. This lifts the two dimensional (2D) image plane to up to five dimensional (5D) spaces (2D translation plus rotation, scaling and reflection). Thanks to the steerable CNNs approach, efficient construction of 3D convolution kernels for translation-rotation G-convolutions is demonstrated in Marcos et al. (2017) and Cheng et al. (2019). It involves stacking rotated versions of 2D kernels to form a 3D kernel, an approach similarly adopted for achieving translation-scale Sosnovik et al. (2020) and translation-rotation-scale equivariance Gao et al. (2022); Sun & Blu (2023). However, this method inherently limits the multi-dimensional kernels to augmented 2D kernels. This restricts the range of learnable kernels to a subset of those possible in G-CNNs. While the multi-dimensional kernel is more general than augmented 2D kernel, it incurs significant computational costs due to the multi-dimensional operations involved.

An alternative approach for achieving continuous equivariance involves mapping elements of the Lie group to the Lie algebra and representing convolution kernels with B-splines Bekkers (2019) (B-splines CNN) and multilayer perceptrons (MLPs) Finzi et al. (2020). This method allows for handling non-uniformly sampled data on arbitrary dimensions, offering significant flexibility. However, there is a trade-off for this flexibility, when it comes to achieving equivariance with respect to the semi-direct product structure, such as the similarity group $Sim(n) \cong \mathbb{R}^n \rtimes H$. Here, convolution kernels are tailored relative to the group $H$ and convolved point-wise with data on the $\mathbb{R}^n$ space. If the data is uniformly sampled, this equates to a discrete convolution, which fails to achieve continuous translation equivariance. Furthermore, the approximation nature of MLPs precludes this method from achieving exact equivariance.

In this paper, we address the challenge of full similarity group equivariant CNNs and hence continuous translating, rotating and scaling 5D kernels in G-CNN by leveraging the steerable and approximately shiftable and scalable basis proposed in Zhang & Williams (2022). The key contributions of this study are summarized as follows:

**Extension and Refinement of Basis Functions:** We have refined and extended the basis function proposed by Zhang & Williams (2022) to include capabilities for reflection, and enhanced its properties to be steerable, approximately shiftable, and scalable.

**Design and Construction of SECNNs:** Zhang & Williams (2022) conducted experiments with a shallow implementation involving only a single convolution with a precomputed kernel. Adopting the steerable CNNs approach, we designed and constructed SECNNs using the enhanced basis function. These networks learn convolutional kernels capable of continuous translation, rotation, and scaling, primarily targeting computer vision tasks involving 2D images.

**Performance and Computational Optimization:** The representation and weight spaces of SECNNs are adjustable to subgroups of the similarity group to cater to different tasks. We have made SECNNs computationally feasible and optimized SECNNs' performance through strategic design choices, such as employing a cropped Fourier series of the basis function and implementing nonlinear and normalization functions in the spatial domain on real-valued representations.

**Empirical Evaluation and Benchmarking:** We conducted a comprehensive benchmark study comparing our SECNNs with state-of-the-art equivariant networks on image datasets such as MNIST variants and CIFAR10/100. This study not only illustrates the empirical advantages of our network but also explores various design choices within our SECNN framework.

Table 1: A comparison of recent group equivariant CNNs with continuous translation (trans), rotation (roto), and scale equivariance. Mark ∘ stands for approximately shiftable or scalable. See Section 1.1 for the reference of each method and Section 2 for the definitions of shiftable, steerable and scalable.

| Equivar. | translation-rotation | | | translation-scale | | translation-rotation-scale | | | |
|---|---|---|---|---|---|---|---|---|---|
| Methods | G-CNN | RotEqNet | H-Net, RotDFC, SFCNN, E2CNN | DSSCNN | SESN | RST-CNN | SREN | B-splines, Separable | SECNN (ours) |
| steerable | ✗ | ✗ | ✓ | ✗ | ✗ | ✓ | ✓ | ∘ | ✓ |
| shiftable | ✗ | ✗ | ✗ | ✗ | ✗ | ✗ | ✗ | ✗ | ∘ |
| scalable | ✗ | ✗ | ✗ | ✗ | ✗ | ✗ | ✓ | ∘ | ∘ |

## 1.1 RELATED WORK

**The Steerable CNNs Approach:** H-Net (Worrall et al. (2017)) employed circular harmonics as a steerable basis for continuous rotation by manipulating harmonic phases. Subsequent research SFCNN (Weiler et al. (2018)), E2CNN (Weiler & Cesa (2019)) and RotDFC (Cheng et al. (2019)) used steerable bases to condense three-dimensional weights to two dimensions, enhancing computational efficiency while retaining the three-dimensional intermediate representations. RotEqNet (Marcos et al. (2017)) proposed a rotation equivariant G-CNN that returns a vector field about orientation. Similarly, Marcos et al. (2018) proposed a scale equivariant G-CNN for vector field. Moreover, SESN (Sosnovik et al. (2020; 2021a;b)) and deep scale-space CNN (DSSCNN, Worrall & Welling (2019)) have led to the creation of translation and scale subgroup equivariant CNNs by applying predefined scale operations on kernels. Likewise, RST-CNN(Gao et al. (2022)) have extended the translation and rotation (or scale) subgroup equivariant CNNs to develop similarity group CNNs with four-dimensional intermediate representations and two-dimensional weights. In a more recent development, SREN(Sun & Blu (2023)) advocate the utilization of a steerable and scalable basis to continuously rotate and scale two-dimensional kernels. This method calculates the optimal orientation and scale in the extra two dimensions, efficiently compressing the four-dimensional intermediate representation to two dimensions. While the aforementioned works leverage predefined bases, which are limited in their generalization to other symmetry transformations, Zhdanov et al. (2024) proposed the use of MLPs to learn steerable kernels (or bases) for translation and other compact groups.

In contrast to these approaches, we derive a basis for the similarity group, which is not a compact group. This basis provides a theoretical framework for achieving exact continuous similarity equivariance. Table 1 offers a comparative summary of these developments in steerable CNNs, contrasting them with our proposed SECNNs.

**The Lie Group and Lie Algebra Approach:** The integration of Lie Algebra, MLPs (and B-splines), and Monte Carlo methods provides a flexible framework for exploring various Lie group equivariances. MacDonald et al. (2022) enhanced this approach to include Lie groups whose exponential maps are not surjective, such as the affine group. Further advancements by Mironenco & Forré (2024) extended this methodology to accommodate Lie groups that are neither compact nor abelian, broadening the applicability of this approach.

One significant challenge with expanding the group to incorporate more transformations is the associated increase in computational costs. To address this, Knigge et al. (2022) introduced separable G-CNNs, which keep the convolution kernels for $\mathbb{R}^n$ and $H$ separable, enhancing computational efficiency. Building on this idea, Bekkers et al. (2023) proposed associating invariant attributes to point pairs, which boosts the computational efficiency of G-CNNs. In a parallel development, Li et al. (2024) achieved affine equivariance by employing differential invariants, although this method diverges from the traditional Lie group and Lie algebra approach. Several methods within this category are categorized as approximated steerable in Table 1 due to their use of B-splines and MLPs.

Most methods of both steerable CNNs approach and this approach treat representations in $\mathbb{R}^n$ as either point clouds or uniformly sampled points, which does not guarantee continuous translation equivariance. Our method specifically addresses this limitation by utilizing shiftable basis functions to ensure continuous translation equivariance effectively

## 2 BACKGROUND

In this section, we begin by providing background on the similarity group and the concept of equivariance. We then demonstrate how group convolution can achieve group equivariance and

introduce the definition of similarity group convolution. Finally, we show that steerable filters and the analytical Fourier-Mellin transform can be utilized to implement continuous transformations within similarity group convolution.

**Similarity Group** Similarity transformations consist of translations, rotations, uniform scaling, and reflections—all of which are shape-preserving transformations. Since 2D similarity transformations form a subset of 2D affine transformations without shearing, the 2D similarity group is a subgroup of the 2D affine group. Consequently, the 2D similarity group can be expressed as a semidirect product:

$$\text{Sim}(2) \cong \{\mathbb{R}^2, +\} \rtimes H, \tag{1}$$

where $\{\mathbb{R}^2, +\}$ is the translation group and $H$ is a group consisting of rotations, uniform scaling, and reflections (with a particular focus on horizontal reflections so as to implement the Dihedral group in this paper). This group has five degrees of freedom: two for translation, one for rotation, one for uniform scaling and one for reflection.

**Group Equivariance** A function $\Phi$ is said to be equivariant to a group $G$ if, whenever the input is transformed by the group action $T$, the output undergoes the same transformation $T'$ (in different function space). Mathematically, this can be expressed as: $\Phi(T(x)) = T'(\Phi(x))$.

**Similarity Group Convolution** Group equivariance can be achieved by group convolution. Kondor & Trivedi (2018) have established that a convolutional structure is not only sufficient but also a necessary condition for achieving equivariance to the action of a compact group. If $G$ is a compact group and $f$ and $g$ are two functions $G \to \mathbb{C}$, then the convolution of $f$ with $g$ is defined as

$$(f * g)(u) = \int_G f(v)g(uv^{-1})d\mu(v), \tag{2}$$

where $\mu$ is the Haar measure.

Now, we aim to define a similarity group convolution, denoted as $*_{\text{sim}(2)}$ (simConv), that achieves similarity group equivariance. Let the variables $u = (\mathbf{x}, \phi, \rho, a)$ and $v = (\mathbf{y}, \theta, r, b)$ of Eq. 2 be elements of $\mathbb{R}^2 \times S^1 \times \mathbb{R}^+ \times \{\pm 1\}$. Note that the similarity group is not a compact group. While Kondor & Trivedi (2018) restricts $G$ to be compact to guarantee a unique measure $\mu$, the similarity group is a subgroup of the affine group, which is locally compact. This means a Haar measure exists and is unique up to a constant multiplier: $\int_{\mathbb{R}^2} d\mathbf{y}$ for the translation group $\{\mathbb{R}^2, +\}$, $\int_{S^1} d\theta/2\pi$ for the rotation group SO(2) and $\int_{\mathbb{R}^+} dr/r$ for the uniform scaling group $\{\mathbb{R}^+, \times\}$. The similarity transformation group $\text{Sim}(2) \cong \{\mathbb{R}^2, +\} \rtimes H$ is the semidirect product of the translation group and the group of rotations, uniform scaling, and horizontal reflections. Its group product is given by:

$$uv^{-1} = (\mathbf{A}^{-1}\mathbf{x} - \mathbf{y}, \phi - \theta, \frac{\rho}{r}, ab), \tag{3}$$

where the transformation matrix $\mathbf{A}$ is defined as: $\mathbf{A} = \begin{bmatrix} br\cos\theta & -br\sin\theta \\ r\sin\theta & r\cos\theta \end{bmatrix} \in \mathbb{R}^{(2\times 2)1}$.

In summary, the simConv $*_{\text{sim}(2)}$ is defined as follows:

$$(f *_{\text{sim}(2)} g)(\mathbf{x}, \phi, \rho, a) = \sum_b \int_{\mathbb{R}^2 \times S^1 \times \mathbb{R}^+} f(\mathbf{y}, \theta, r, b)g(\mathbf{A}^{-1}\mathbf{x} - \mathbf{y}, \phi - \theta, \frac{\rho}{r}, ab)d\mathbf{y}\frac{d\theta}{2\pi}\frac{dr}{r}. \tag{4}$$

**Shiftable, Steerable and Scalable Filters** Steerable filters have been widely used to implement continuous rotations and discrete translations of the function $g$ with respect to the linear operation $\mathbf{A}^{-1}\mathbf{x} - \mathbf{y}$ of the group product in Eq. 3 (see Section 1.1). According to Freeman et al. (1991), a kernel $g$ is *steerable* if and only if, for arbitrary orientation $\theta \in S^1$ and a set of rotated copies of $g$, $\{g_{\theta_i} | i \in \mathbb{Z}, \theta_i \in S^1\}$, there exists a set of coefficients $\{c_i(\theta) \in \mathbb{C} | i \in \mathbb{Z}\}$ such that

$$f * g_\theta = \sum_i c_i(\theta)(f * g_{\theta_i}), \tag{5}$$

where the coefficients $c_i(\theta)$ are independent of $f$ (see Fig. 1 (a)). Essentially, a steerable filter can be understood as a band-limited Fourier series of functions for the orientation axis of the polar coordinate

---

[1]Horizontal reflection is performed before rotation to achieve the Dihedral group action.

system. Simoncelli et al. (1992) broadened this concept to aperiodic domain such as position and scale. A *shiftable* filter is a band-limited Fourier series of functions for translation. Since rotation and scaling are translations in the log-polar coordinate system, a shiftable filter defined in this coordinate system is named steerable and/or *scalable* filter. In recent literature, these filters—shiftable, steerable, and scalable—are collectively known as steerable filters w.r.t. translation, rotation, and scaling (Cohen & Welling (2017)).

**Analytical Fourier-Mellin Transform** The basis of the Analytical Fourier-Mellin Transform (AFMT) (Ghorbel, 1994) is a complete basis. Its basis functions can be made steerable and approximately shiftable and scalable. Using these basis functions, an approximately continuous translation, rotation and scaling corresponding to the linear operation $\boldsymbol{A}^{-1}\boldsymbol{x} - \boldsymbol{y}$ (Eq. 4) in the similarity group convolution can be implemented. Consequently, similarity group convolutional networks employing this implementation learn filters that are steerable and approximately shiftable and scalable.

The AFMT is formulated through a combination of the Fourier transform of orientation and the bilateral Laplace transform of log-scale in the log-polar coordinate system. Consider $(x_0, x_1)$ as the 2D Cartesian coordinates and let $\phi = \arctan\frac{x_1}{x_0}$ and $\rho = \sqrt{x_0^2 + x_1^2}$. $(\phi, \rho)$ are the axes defining the polar coordinate system, whose origin is at $(x_0, x_1) = (0, 0)$, and $(\omega_\phi, \alpha_\rho + i\omega_\rho)$ representing the frequencies in the AFMT. The frequency pair consists of Fourier frequency ($\omega_\phi \in \mathbb{Z}$) for orientation $\phi$ and Laplacian frequency ($(\alpha_\rho + i\omega_\rho) \in \mathbb{C}$) for log-scale $\log\rho$. For a two-dimensional function $g(\phi, \rho) : \mathbb{R}^2 \to \mathbb{C}$ in this system, the AFMT and its inverse are defined as follows:

$$\mathcal{M}\{g\}(\omega_\phi, \alpha_\rho + i\omega_\rho) = \int_0^\infty \int_0^{2\pi} g(\phi, \rho) e^{-i\omega_\phi\phi} \rho^{-(\alpha_\rho + i\omega_\rho)} \frac{d\phi}{2\pi} \frac{d\rho}{\rho}, \tag{6}$$

$$\mathcal{M}^{-1}\{G\}(\phi, \rho) = \int_{-\infty}^\infty \sum_{\omega_\phi = -\infty}^\infty G(\omega_\phi, \alpha_\rho, \omega_\rho) e^{i\omega_\phi\phi} \rho^{\alpha_\rho + i\omega_\rho} \, d\omega_\rho. \tag{7}$$

In the rest of this paper, we denote the AFMT basis functions as

$$p_{(\omega_\phi, \alpha_\rho + i\omega_\rho)}(\phi, \rho) = e^{i\omega_\phi\phi} \rho^{\alpha_\rho + i\omega_\rho}. \tag{8}$$

# 3 EFFICIENT CONTINUOUS IMPLEMENTATION OF SIMILARITY GROUP CONVOLUTION

In this section, we aim to approximate the group convolution Eq. 4 through a basis expansion (similar to Eq. 5 but generalized to the similarity group, see Fig. 1 (b)) to 1) avoid directly sampling the similarity group, which does not achieve full *continuous* equivariance, and 2) avoid direct interpolation, which introduces errors.

To achieve this, we first modify the 2D steerable AFMT basis functions defined in Eq. 8 to be approximately shiftable and scalable; we call this the *S3 basis*. This modification allows continuous translations, rotations and scalings of a kernel to be represented as linear combinations of a finite set of modified basis functions. Note that these transformations correspond to the linear operation $\boldsymbol{A}^{-1}\boldsymbol{x} - \boldsymbol{y}$ on the $\mathbb{R}^2$ plane in Eq. 3. Then, to account for the rest operations $(\phi - \theta, \frac{\rho}{r}, ab)$ on the $S^1 \times \mathbb{R}^+ \times \{\pm 1\}$ space, we generalize these modified basis functions to higher-dimensional basis functions defined over the 5D function space $\mathbb{R}^2 \times S^1 \times \mathbb{R}^+ \times \{\pm 1\} \to \mathbb{C}$. This generalization enables us to incorporate the full range of transformations in the similarity group. Finally, we implement the similarity group convolution by utilizing these generalized and modified basis functions.

## 3.1 CONSTRUCTION OF 5D STEERABLE, SHIFTABLE AND SCALABLE BASIS FUNCTIONS

In this subsection, we construct steerable and approximately shiftable and scalable basis functions in the 5D function space $\mathbb{R}^2 \times S^1 \times \mathbb{R}^+ \times \{\pm 1\} \to \mathbb{C}$ from the 2D steerable basis functions $p_{(\omega_\phi, \alpha_\rho + i\omega_\rho)} : \mathbb{R}^2 \to \mathbb{C}$ defined in Eq. 8.

### 3.1.1 MODIFICATION OF THE 2D AFMT BASIS FUNCTIONS TO BE APPROXIMATELY SHIFTABLE AND SCALABLE.

The modification involves three steps. First, we derive the Fourier transform of the 2D basis functions $p_{(\omega_\phi, \alpha_\rho + i\omega_\rho)} : \mathbb{R}^2 \to \mathbb{C}$ to obtain their spatial frequency representations. Second, we assume

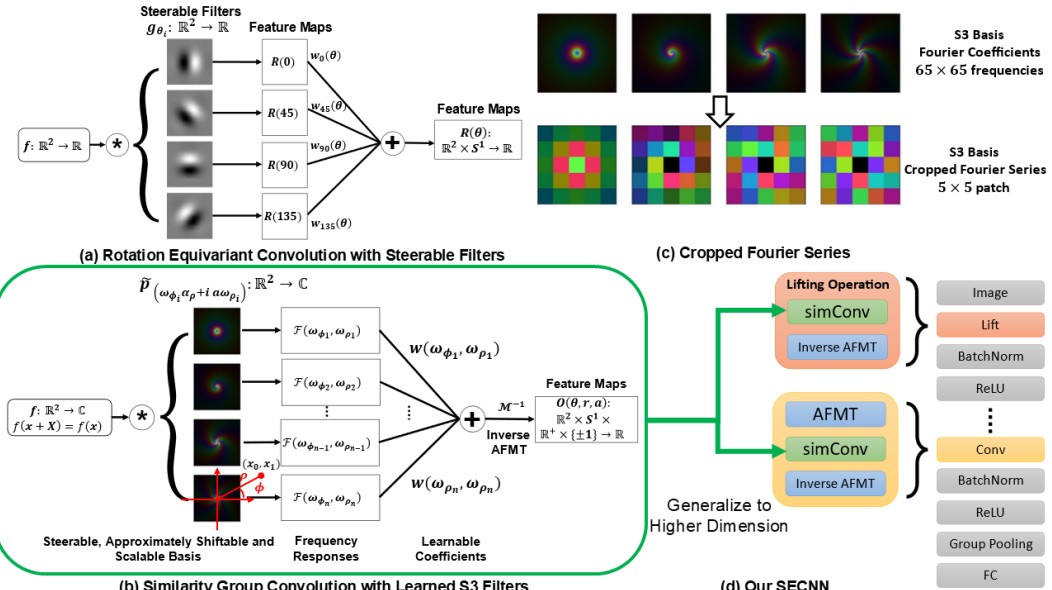

Figure 1: System Diagram of SECNN. (a) Steerable filter approach: The feature map $R(\theta)$ is a linear combination of four feature maps. There is no need to explicitly rotate the kernel $g$. (b) Proposed S3 basis approach: The feature map $O(\theta, r, a)$ is the output of inverse AFMT applied to a linear combination of frequency responses. The learned filters are S3 filters, and no explicit transformations are required. (c) Visualization of the cropped Fourier series $\tilde{p}$: The phase of complex values are mapped to hue with red indicating positive real. (d) Integration into Neural Networks: The two simConv layers can readily replace conventional convolutional layers. The orange block corresponds to the simConv layer visualized in (b) (see Eq. 14 and 16), while the yellow block represents a generalization of (b), expanding the function space to 4D or 5D (see Eq.15 and 17).

periodicity in both position and scale to modify these spatial frequency representations, allowing us to work with integer frequencies and construct Fourier series using finite and discrete frequency components. Third, we crop the modified basis functions in both the frequency and spatial domains to enhance computational efficiency.

**Fourier Transform and Fourier Coefficients:** When the basis functions $p_{(\omega_\phi, \alpha_\rho + i\omega_\rho)} : \mathbb{R}^2 \to \mathbb{C}$ defined in Eq. 8 are represented in the Cartesian coordinate system, and let $\boldsymbol{x} = (x_0, x_1) \in \mathbb{R}^2$ symbolize Cartesian coordinates, the associated polar coordinates, with the origin at $(0,0) \in \mathbb{R}^2$, are represented as $\phi = \arctan \frac{x_1}{x_0}$ and $\rho = \sqrt{x_0^2 + x_1^2}$. There exist an analytical Fourier transform solution of this function when $-2 < \alpha_\rho < -0.5$ (Zhang & Williams, 2022). Interested readers may find derivation details in Appendix A. It is simple to modify the solution to create the Fourier coefficients by multiplying a $2\pi/X$ and $2\pi/S$ term with the spatial frequencies $(\omega_{x_0}, \omega_{x_1})$ and scale frequency $\omega_\rho$ respectively, where $X$ is the spatial period and $S$ is the scale period. The Fourier coefficients are defined as:

$$P(\omega_{\boldsymbol{x}}, \omega_\phi, \alpha_\rho + i\omega_\rho) = \pi(-i)^{|\omega_\phi|} e^{j\omega_\phi \bar{\phi}} \left(\frac{X}{\pi \bar{\rho}}\right)^{2 + \alpha_\rho + i\frac{2\pi}{S}\omega_\rho} \frac{\Gamma\left(\frac{1}{2}(2 + |\omega_\phi| + \alpha_\rho + i\frac{2\pi}{S}\omega_\rho)\right)}{\Gamma\left(\frac{1}{2}(|\omega_\phi| - (\alpha_\rho + i\frac{2\pi}{S}\omega_\rho))\right)}, \quad (9)$$

where $\Gamma$ represents the Gamma function, $\omega_{\boldsymbol{x}} = (\omega_{x_0}, \omega_{x_1})$, $\bar{\phi} = \arctan \frac{\omega_{x_1}}{\omega_{x_0}}$ and $\bar{\rho} = \sqrt{\omega_{x_0}^2 + \omega_{x_1}^2}$. All the frequencies $\omega_{\boldsymbol{x}}, \omega_\phi$ and $\omega_\rho$ are integer frequencies. A simplified representation of Eq. 9 is

$$P(\omega_{\boldsymbol{x}}, \omega_\phi, \alpha_\rho + i\omega_\rho) = Z(\omega_\phi, \alpha_\rho + i\omega_\rho) e^{j\omega_\phi \bar{\phi}} \bar{\rho}^{-(2 + \alpha_\rho + i\frac{2\pi}{S}\omega_\rho)}$$

where $Z(\omega_\phi, \alpha_\rho + i\omega_\rho) \in \mathbb{C}$ is a constant with respect to $\omega_{\boldsymbol{x}}$.

**Approximately Steerable and Scalable:** When $-2 < \alpha_\rho < -0.5$ the polynomial decay functions $\rho^{\alpha_\rho}$ of Eq. 8 and $\bar{\rho}^{-(2+\alpha_\rho)}$ of Eq. 9 form joint localization in both the spatial and frequency domain. Note that the localization resulted from the polynomial decay functions is not band-limiting.

Therefore, the Fourier series constructed using a finite number of spatial and scale frequencies are steerable and approximately shiftable and scalable.

**Cropping Fourier Series in the Frequency and Spatial Domain:** We want to improve the computational efficiency of the simConv defined in Eq. 4. This is done by spatially cropping the Fourier series computed from the Fourier coefficients defined in Eq. 9,

$$\tilde{p}_{(\omega_\phi, \alpha_\rho + i\omega_\rho)}(\boldsymbol{x}) = \sum_{\omega_{\boldsymbol{x}}} P(\omega_{\boldsymbol{x}}, \omega_\phi, \alpha_\rho + i\omega_\rho)e^{i\omega_{\boldsymbol{x}}\boldsymbol{x}}, \tag{10}$$

to the $5 \times 5$ kernel size ($x_0, x_1 \in [-2, 2]$) used in this paper (see Fig. 1 (c)). This not only drastically reduces the computational cost but also allows us to precompute the basis (Eq. 10) with large enough spatial frequencies, for instance $1024 \times 1024$ ($\omega_{x_0}, \omega_{x_1} \in [-512, 511]$), to avoid aliasing.

### 3.1.2 GENERALIZATION OF 2D BASIS FUNCTIONS TO 5D

**Horizontal Reflection of the Basis Function:** We generalize the basis functions to include horizontal reflections to achieve the Dihedral group action. Let $a \in \{\pm 1\}$ be the reflection indicator, the reflected basis function (Eq. 8) is defined as:

$$p_{(\omega_\phi, \alpha_\rho + i\omega_\rho)}(\phi, \rho, a) = e^{ia\omega_\phi\phi}\rho^{\alpha_\rho + i\omega_\rho}. \tag{11}$$

The reflection indicator $a$ affects only the orientation frequency, as a horizontally flipped basis function has a negative orientation frequency of the original version. Accordingly, the reflected and cropped Fourier series (Eq. 10) is defined as:

$$\tilde{p}_{(\omega_\phi, \alpha_\rho + i\omega_\rho)}(\boldsymbol{x}, a) = \sum_{\omega_{\boldsymbol{x}}} P(\omega_{\boldsymbol{x}}, a\omega_\phi, \alpha_\rho + i\omega_\rho)e^{i\omega_{\boldsymbol{x}}\boldsymbol{x}}, \tag{12}$$

**5D Basis Functions:** To simplify, let's exclude the translation component $-\boldsymbol{y}$ in Eq. 3 and define: $u = (x_0, x_1, \theta, r, a) = (\phi, \rho, \theta, r, a)$ and $v = (0, 0, \Delta\theta, \gamma, b)$, where $(\phi, \rho)$ are the polar coordinates corresponding to $\boldsymbol{x} = (x_0, x_1)$ for origin at $(0, 0)$. The group product in Eq. 3 then becomes $uv^{-1} = (\phi - \Delta\theta, \rho/\gamma, \theta - \Delta\theta, r/\gamma, ab)$. Notice that the pairs $(\phi, \theta)$ and $(\rho, r)$ are not orthogonal axes. To obtain orthogonal axes, we consider the transformation to: $(\phi, \theta - \phi, \rho, \frac{r}{\rho}, a)$. This leads us to define the orthogonal basis functions as:

$$
\begin{aligned}
p_{(\omega_\phi, s_\rho, \omega_\theta, s_r)}(\phi, \rho, \theta, r, a) &= \rho^{s_\rho}e^{ia\omega_\phi\phi}\left(\frac{r}{\rho}\right)^{s_r}e^{ia\omega_\theta(\theta - \phi)} \\
&= p_{(\omega_\phi - \omega_\theta, s_\rho - s_r)}(\phi, \rho, a)p_{(\omega_\theta, s_r)}(\theta, r, a) \\
&\approx \tilde{p}_{(\omega_\phi - \omega_\theta, s_\rho - s_r)}(\boldsymbol{x}, a)p_{(\omega_\theta, s_r)}(\theta, r, a),
\end{aligned}
\tag{13}
$$

where $s_\rho = \alpha_\rho + i\omega_\rho$, $s_r = \alpha_r + i\omega_r$ and $(\omega_\phi, s_\rho, \omega_\theta, s_r)$ are the frequencies for $(\phi, \rho, \theta, r)$. The two factors are two 2D basis functions defined in Eq. 11. As the factor $p_{(\omega_\phi - \omega_\theta, s_\rho - s_r)}(\phi, \rho, a) = p_{(\omega_\phi - \omega_\theta, s_\rho - s_r)}(\boldsymbol{x}, a)$, we replace it with the cropped Fourier series $\tilde{p}$ defined in Eq. 12. The change of coordinate system from $(\phi, \rho)$ to $\boldsymbol{x} = (x_0, x_1)$ is visualized in Fig. 1 (b).

### 3.2 IMPLEMENTATION OF SIMCONV USING THE 5D S3 BASIS FUNCTIONS

In this section, we utilize the cropped Fourier Series (Eq. 10 and 13) to implement similarity group convolutions for two specific groups: a continuous subgroup consisting of translation, rotation and scaling, and a discrete similarity group. Specifically, the group $H$ in Eq. 1 for these two cases are $SO(2) \times \{\mathbb{R}^+, \times\}$ and $D_n \times \{\mathbb{R}^+, \times\}$, respectively.

The continuous translation, rotation and scaling subgroup means that we can model its group action by using finite and discrete representations, *e.g.,* integer frequency responses. The discrete similarity group is discrete as the Dihedral group $D_n$ (rotation-reflection group) is discrete. This means that we have to work on a finite number of orientations. Otherwise, the Dihedral group elements are infinite and we cannot use discrete and finite representations to achieve its group action.

**Lifting Operation:** The group elements of the two groups are either 4D[2] or 5D (see Eq. 3), while the input images we are dealing with are 2D functions. Therefore, we need a lifting operation that lifts 2D functions to 5D functions.

---

[2]For groups without reflection, we can simply assume that the reflection indicator is a constant 1.

For 2D images $\mathbb{R}^2 \to \mathbb{R}$, we assume that their orientation to be 0, scale to be 1 and reflection indicator to be 1. Additionally, we denote the AFMT operation as $\mathcal{M} : S^1 \times \mathbb{R}^+ \to \mathbb{C}$ to compute the frequency domain representations in $S^1 \times \mathbb{R}^+$ and $s_\rho = \alpha_\rho + i\omega_\rho$. For a 2D input $f'$ and kernel $g'$, the frequency domain representation of $g'$ is $G'(\omega_\phi, s_\rho) = \mathcal{M}\{g'\}(\omega_\phi, s_\rho)$. The simConv for the continuous subgroup is defined as:

$$\mathcal{M}\left\{f' *_{\text{sim}(2)} g'\right\}_{(-\omega_\phi, -s_\rho)}(\boldsymbol{x}) = \left\{f' * \tilde{p}_{(\omega_\phi, s_\rho)}\right\}(\boldsymbol{x})G'(\omega_\phi, s_\rho), \tag{14}$$

and the simConv for the discrete similarity group is defined as:

$$\left(f' *_{\text{sim}(2)} g'\right)(\boldsymbol{x}, \theta, r, a) = \sum_{\omega_\phi, \omega_\rho} \left(f' * \tilde{p}_{(\omega_\phi, s_\rho)}\right)(\boldsymbol{x}, a)G'(\omega_\phi, s_\rho)e^{ia\omega_\phi\theta}r^{\alpha_\rho + i\omega_\rho}. \tag{15}$$

The main difference between Eq. 14 and 15 is that the former results in frequency responses regarding $S^1 \times \mathbb{R}^+$ whereas the latter utilizes an inverse AFMT (with flipped inverse basis function $e^{ia\omega_\phi\theta}$) to convert the frequency responses to discrete orientations and scales. The detailed derivations are in Appendix B.1.

**Implementation of the simConv:** For input $f$ and kernel $g$ in $\mathbb{R}^2 \times S^1 \times \mathbb{R}^+ \to \mathbb{R}$, the frequency domain representations are denoted as $F_{(\omega_\phi, s_\rho)}(\boldsymbol{x}) = \mathcal{M}\{f\}_{(\omega_\phi, s_\rho)}(\boldsymbol{x})$ and $G(\omega_\phi, s_\rho, \omega_\theta, s_r)$ respectively. The simConv for the continuous subgroup is defined as:

$$\mathcal{M}\{f *_{\text{sim}(2)} g\}_{(\omega_\theta, s_r)}(\boldsymbol{x}) = \sum_{\omega_\phi, \omega_\rho} \left\{F_{(\omega_\phi, s_\rho)} * \tilde{p}_{(\omega_\phi - \omega_\theta, s_\rho - s_r)}\right\}(\boldsymbol{x})G(\omega_\phi, s_\rho, \omega_\theta, s_r). \tag{16}$$

For input $f$ and kernel $g$ in $\mathbb{R}^2 \times S^1 \times \mathbb{R}^+ \times \{\pm 1\} \to \mathbb{R}$, the frequency domain representations are denoted as $F_{(\omega_\phi, s_\rho)}(\boldsymbol{x}, a) = \mathcal{M}\{f\}_{(\omega_\phi, s_\rho)}(\boldsymbol{x}, a)$ and $G(\omega_\phi, s_\rho, \omega_\theta, s_r, b)$ respectively. The simConv for the discrete similarity group is defined as:

$$(f *_{\text{sim}(2)} g)(\boldsymbol{x}, \theta, r, b) =$$

$$\sum_{\omega_\theta, \omega_r} \left( \sum_{\omega_\phi, \omega_\rho, a} \left\{F_{(\omega_\phi, s_\rho)}(a) * \tilde{p}_{(\omega_\phi - \omega_\theta, s_\rho - s_r, a)}\right\}(\boldsymbol{x})G(\omega_\phi, s_\rho, \omega_\theta, s_r, ab) \right) e^{ib\omega_\theta\theta}r^{\alpha_r + i\omega_r} \tag{17}$$

The detailed derivations are in Appendix B.2. It is worth noting that the AfMT coefficients $G'$ and $G$ of the convolution kernels $g'$ and $g$ represent learnable parameters. Hence, there exists no explicit requirement to perform AFMT on these kernels.

# 4 CONSTRUCTION OF DEEP SIMILARITY GROUP EQUIVARIANT CNNS

In this section, we discuss deep SECNN architecture details, such as nonlinear function, normalization, pooling and number of channels, that help achieve optimal performance. A typical feed-forward architecture for SECNNs is illustrated in Fig. 1 (d).

**Nonlinearity and Normalization**: Feature maps are converted back and forth between the spatial domain ($S^1 \times \mathbb{R}^+$) and frequency domain even if the simConv is for the continuous subgroup. This is because the conversions allow us to leverage the spatial domain and real number ReLU and BatchNorm. We avoid the same operations for complex numbers because maintaining equivariance requires preserving the phases of complex numbers, which contain essential information about orientation and scale. Therefore, the nonlinear activation and normalization functions are designed to only affect the magnitudes of complex numbers. However, our experiments show that this frequency domain implementation consistently yields sub-optimal results. As stated in Trabelsi et al. (2017), the non-holomorphic nature of ReLU on magnitudes of complex numbers, which does not satisfy the Cauchy-Riemann equations, might be the underlying reason.

**Group Pooling**: In our SECNN architecture, a group pooling layer is strategically placed after the last convolution layer to aggregate feature maps in $\mathbb{R}^2 \times S^1 \times \mathbb{R}^+ \times \{\pm 1\}$. The output from this pooling layer then feeds into the final fully-connected layer. The choice of pooling operation is tailored to the specific requirements of the application. For instance, global max pooling is employed for invariant classification tasks, which demand a high level of feature stability regardless of geometric transformations. Conversely, for natural image classification, we utilize an orientation histogram approach that better captures the variability inherent in natural scenes.

Table 2: Translated, Rotated and Scaled MNIST Classification Error Rates (%). O, T, R and S stand for original, translated, rotated and scaled MNIST respectively. The concatenated letters mean that it is the mixture of transformations. Mean and standard deviation are computed from 5 random seed variations. See Table 4 in Appendix for the network illustration of secnn-3D, 4D and mix

| | # params | O | R | TR | S | TS | RS | TRS |
|---|---|---|---|---|---|---|---|---|
| CNN | 2.54M | $0.62_{\pm 0.08}$ | $1.03_{\pm 0.01}$ | $2.88_{\pm 0.07}$ | $1.16_{\pm 0.07}$ | $1.43_{\pm 0.06}$ | $2.37_{\pm 0.11}$ | $4.78_{\pm 0.10}$ |
| SESN | 2.37M | $0.62_{\pm 0.02}$ | $1.28_{\pm 0.02}$ | $3.00_{\pm 0.06}$ | $1.24_{\pm 0.05}$ | $1.57_{\pm 0.08}$ | $2.88_{\pm 0.08}$ | $5.25_{\pm 0.05}$ |
| E2CNN | 2.69M | $\mathbf{0.47_{\pm 0.04}}$ | $0.78_{\pm 0.03}$ | $3.10_{\pm 0.07}$ | $1.09_{\pm 0.06}$ | $1.76_{\pm 0.07}$ | $1.70_{\pm 0.05}$ | $5.07_{\pm 0.10}$ |
| sim2CNN | 0.79M | $0.53$ | $\mathbf{0.59_{\pm 0.008}}$ | $2.52$ | $1.13$ | $1.50$ | $1.87$ | $4.46$ |
| secnn$_{3D}$ | 2.52M | $0.50_{\pm 0.02}$ | $0.84_{\pm 0.02}$ | $\mathbf{2.41_{\pm 0.03}}$ | $0.95_{\pm 0.04}$ | $1.19_{\pm 0.05}$ | $\mathbf{1.50_{\pm 0.07}}$ | $4.08_{\pm 0.05}$ |
| secnn$_{4D}$ | 2.57M | $0.56_{\pm 0.04}$ | $0.95_{\pm 0.03}$ | $2.61_{\pm 0.07}$ | $1.00_{\pm 0.03}$ | $1.26_{\pm 0.03}$ | $1.69_{\pm 0.05}$ | $4.10_{\pm 0.13}$ |
| secnn$_{mix}$ | 2.54M | $0.50_{\pm 0.03}$ | $0.85_{\pm 0.05}$ | $2.47_{\pm 0.02}$ | $\mathbf{0.92_{\pm 0.02}}$ | $\mathbf{1.16_{\pm 0.06}}$ | $1.51_{\pm 0.07}$ | $\mathbf{3.98_{\pm 0.09}}$ |

The orientation histogram is computed as follows: Feature maps in $\mathbb{R}^2 \times S^1 \times \mathbb{R}^+ \times \{\pm 1\}$ are first max-pooled over the scale space $\mathbb{R}^+$, resulting in $\mathbb{R}^2 \times S^1 \times \{\pm 1\}$. For each sparse position in $\mathbb{R}^2$, the orientation with the maximum value is identified. The feature maps are then divided into $m \times m$ cells, and an orientation histogram is computed for each cell. Finally, all histograms in $\{\pm 1\}$ and the $m \times m$ cells are concatenated. This method is similar to the SIFT descriptor (Lowe (2004)). See Appendix C.1 for more disscusion.

**Trading Off Full Equivariance for More Features and Hence Accuracy**: Previous studies on group equivariant networks often highlight parameter efficiency, demonstrating superior performance over baseline models with a comparable number of parameters. However, despite the generality of weights in $\mathbb{R}^2 \times S^1 \times \mathbb{R}^+ (\times \{\pm 1\})$, the inherently 4D (or 5D) nature of these weights consumes a significant number of parameters. This high parameter count implies that, under a fixed total parameter budget, a SECNN might have fewer channels in each layer compared to traditional models, potentially reducing the diversity of features it can learn. Table 4 illustrates the architectures of SECNNs, where the 4D SECNN has approximately one-third the channels of its baseline counterpart.

To mitigate this issue, we explore methods to increase the number of output channels by reducing the dimensions of weights in certain layers. One approach involves fixing the weight across certain dimensions, rendering it a rectangular function in the frequency domain and a Sinc function in the spatial domain. For example, setting $G(\omega_\phi, s_\rho, \omega_\theta, s_r, ab) = \tilde{G}(\omega_\phi, s_\rho, \omega_\theta)$ implies the weight in the $\mathbb{R}^+$ space is a Sinc function, which preserves the feature maps' dimensions. This configuration equals to using separable kernels (Knigge et al. (2022)) and is utilized in our invariant MNIST classification experiments, where the preservation of feature map dimensions is crucial for achieving equivariance.

Alternatively, the reduction in parameter count per channel can be achieved by applying max-pooling to feature maps followed by lifting them back using Eq. 15, where the weight becomes 2D. This approach has been shown by Sosnovik et al. (2020) to yield optimal results in natural image classification experiments, particularly evident in our CIFAR classification tests.

## 5 EXPERIMENTS

In this section, we conduct two experiments to study and demonstrate the performance of SECNNs. The network and training parameters can be found in Appendix D.3.

**Translated, Rotated and Scaled MNIST** (Dataset details in Appendix D.1) We tested our proposed networks with training-time augmentations, aligning with methodologies employed in previous works E2CNNWeiler et al. (2018); Weiler & Cesa (2019); Sosnovik et al. (2020); Knigge et al. (2022). For our experimental setup, we adopted a six-layer convolutional architecture similar to that used by Weiler & Cesa (2019), which demonstrated state-of-the-art performance on rotated MNIST. Baseline comparisons were drawn against a standard CNN, the rotation-equivariant (discrete translation and $C_{16}$ group) E2CNN (Weiler & Cesa, 2019), the scale-equivariant (discrete translation-scale group) SESN (Sosnovik et al., 2020), and the (discrete translation and continuous rotation-scale group) sim2CNN Knigge et al. (2022) with channels scaled to match 2.5 million parameters across models. We didn't match the parameter count for sim2CNN as it is currently the SOTA on rotated MNIST classification. We simply run its released codes on the datasets we created. We tested SECNNs that involved continuous translation, rotation and scaling group convolutions (Eq. 14 and 16). Their

Table 3: CIFAR10/100 Classification Error Rates (%). SECNN$n$ refers to $n \times n$ group pooling grid.

|  | # params | cifar10 | cifar100 |
|---|---|---|---|
| WRN | 11M | $4.3 \pm 0.12$ | $20.7 \pm 0.24$ |
| SESN | 11M | $3.92 \pm 0.11$ | $19.88 \pm 0.28$ |
| E2CNN | 12M | $3.90 \pm 0.15$ | $\mathbf{18.79 \pm 0.38}$ |
| SECNN1 | 11M | $3.81 \pm 0.02$ | $20.37 \pm 0.06$ |
| SECNN2 | 11M | $\mathbf{3.72 \pm 0.15}$ | $20.17 \pm 0.35$ |

architectures are listed in Table 4 in Appendix C.2. These SECNNs were trained on 4 Nvidia A100 GPUs, utilizing approximately 36 GB of memory, completing 200 epochs in about 38 minutes.

As indicated in Table 2, both E2CNN and Sim2CNN struggled significantly with the combination of translation and rotation. SECNNs excelled on datasets beyond the original and rotated MNIST, demonstrating superior robustness to translations thanks to the shiftable properties of the proposed basis. Interestingly, the results indicate that SECNN-4D performed worse than both SECNN-3D and -Mix, suggesting that while 4D weights provide greater generality, the number of channels also plays a critical role (as it's discussed in Section 4). The SECNN-Mix, which blends weights across different spaces, struck an optimal balance between weight generality and diversity, achieving the best performance on datasets involving translations, rotations, and scaling.

**CIFAR Natural Image Classification** (Dataset details in Appendix D.2) We opted for the Wide Residual Networks (WRN, Zagoruyko & Komodakis (2016)), particularly the WRN16-8 model, as our baseline architecture, aligning with the choices in (discrete translation-scale equivariant) SESN-B Sosnovik et al. (2020) and (discrete translation-Dihedral equivariant) E2CNN-D8D4D4 Weiler & Cesa (2019). A dropout layer with a probability of 0.3 was utilized. For the SECNNs, we chose the ones that are equivariant to the discrete similarity group (which consists of Dihedral group and continuous translation and scaling group). These networks involved the group convolutions defined in Eq. 15 and 17, and their architectures are listed in Table 4 in Appendix C.2. Additionally, we experimented with two cell sizes for the orientation histogram in the group pooling layer, labeled SECNN1 and SECNN2. These networks were trained on 4 Nvidia A100 GPUs, utilizing approximately 154 GB of memory and completing 200 epochs in about 20 hours.

As shown in Table 3, SECNNs outperformed other models on CIFAR10 and ranked third on CI-FAR100, surpassing the performance of the conventional WRN. The slightly weaker performance on CIFAR100 can likely be attributed to the limited number of channels, which may restrict feature diversity crucial for classifying the broader array of categories. Additionally, to maintain a comparable number of parameters and initial channels, the widen-factor for the SECNN-WRN was set to 1, effectively transforming the WRN into a standard ResNet He et al. (2016). While WRNs typically benefit from a broader network, SECNNs faced constraints due to parameter limitations, affecting their potential to leverage wider networks for enhanced performance.

## 6 CONCLUSIONS AND LIMITATIONS

In this paper, we construct SECNNs that are equivariant to a continuous translation, rotation and scaling group and a discrete similarity group. The networks are built on our proposed steerable and approximately shiftable and scalable basis. The former achieved the state-of-the-art results on the classification of translated, rotated and scaled MNIST, demonstrating the effectiveness of the approximate shiftable and scalable properties. Furthermore, the latter achieves the accuracy of other leading group equivariant networks on CIFAR10/100, showcasing the effectiveness of discrete similarity group equivariance.

The primary limitation of our approach is the high computational cost associated with implementing the simConv (see runtime in Table 5 of Appendix D.4). Currently, simConv, as defined in Eq. 16, is similar to a 4D discrete convolution in terms of computational cost. We employed PyTorch's unfold function to implement simConv, which uses a sliding window method that explicitly creates all windows, leading to substantial memory consumption. Optimizing this implementation through a customized CUDA kernel could significantly reduce both computational costs and memory requirements.

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

## A Fourier Transform of the AFMT Basis Functions

In this section, we delve into the Fourier transform of the AFMT basis functions. Initially, section A.1 outlines the methodology for performing a Fourier transform in a polar coordinate representation. Subsequently, section A.2 proceeds to derive the Fourier transform specifically for the basis function.

### A.1 Fourier Transform in a Polar Coordinate Representation

Let $(x, y) \in \mathbb{R}^2$ represent the Cartesian coordinates, and let $u, v \in \mathbb{R}$ be the spatial frequencies corresponding to the $x$- and $y$-dimensions, respectively. Given a function $f' : \mathbb{R}^2 \to \mathbb{C}$, the two-dimensional Fourier transform, denoted as $\mathcal{F}$, is defined by the formula:

$$\mathcal{F}\{f\}(u, v) = \int_{-\infty}^{\infty} \int_{-\infty}^{\infty} f(x, y) e^{-i2\pi(ux+vy)} \, dx \, dy. \tag{18}$$

The polar coordinates in the spatial domain and the frequency domain are $\rho = \sqrt{x^2 + y^2}, \phi = \arctan \frac{y}{x}$ and $\bar{\rho} = \sqrt{u^2 + v^2}, \bar{\phi} = \arctan \frac{v}{u}$ respectively. Considering these definitions, the expression for the product of spatial frequencies $u, v$ and coordinates $x, y$ can be transformed into polar coordinates as follows:

$$\begin{aligned} ux + vy &= \rho\bar{\rho}(\cos\phi\cos\bar{\phi} + \sin\phi\sin\bar{\phi}) \\ &= \rho\bar{\rho}\cos(\phi - \bar{\phi}). \end{aligned} \tag{19}$$

Given this relationship, and considering that the differential area in polar coordinates is expressed as $\rho, d\rho, d\phi$, the two-dimensional Fourier transform of a function $f$ in polar coordinates can be expressed as:

$$\mathcal{F}\{f\}(\bar{\phi}, \bar{\rho}) = \int_0^{2\pi} \int_0^{\infty} f(\rho, \phi) e^{-i2\pi\rho\bar{\rho}\cos(\phi-\bar{\phi})} \rho \, d\rho \, d\phi. \tag{20}$$

### A.2 Fourier Transform of the AFMT Basis Functions

Let $\mathbb{Z}$ represent the space of integers, $\omega_\phi \in \mathbb{Z}$ denote the angular frequency, and $s_\rho = \alpha_\rho + i\omega_\rho \in \mathbb{C}$ denote the radial frequency. Additionally, let $a \in \{-1, 1\}$ be an indicator for reflection. Given these parameters, a basis function can be defined as:

$$p_{(\omega_\phi, \alpha_\rho+i\omega_\rho)}(\phi, \rho, a) = e^{ia\omega_\phi\phi} \rho^{\alpha_\rho+i\omega_\rho}. \tag{21}$$

Given an orientation $\phi$, the effective angular frequency is simplified to $\omega_\rho$ as we assume $a = 1$ for the purposes of this derivation. The Fourier transform of the basis function of AMFT, when expressed in polar coordinates, can be formulated based on Eq. equation 20.

$$\begin{aligned} \mathcal{F}\{p\}_{(\omega_\phi, \alpha_\rho+i\omega_\rho)}(\bar{\phi}, \bar{\rho}) &= \int_0^{2\pi} \int_0^{\infty} \rho^{\alpha_\rho+i\omega_\rho} e^{i\omega_\phi\phi} e^{-i2\pi\rho\bar{\rho}\cos(\phi-\bar{\phi})} \rho \, d\rho \, d\phi \\ &= \int_0^{\infty} \rho^{\alpha_\rho+i\omega_\rho} \int_0^{2\pi} e^{-i\left(2\pi\rho\bar{\rho}\cos(\phi-\bar{\phi})-\omega_\phi\phi\right)} \, d\phi\rho \, d\rho \end{aligned} \tag{22}$$

To compute the Fourier transform of the basis functions as expressed in Eq. equation 22, it is essential to employ Bessel's integral. Consider $x \in \mathbb{R}$ as the variable and $n \in \mathbb{Z}$ as the order of Bessel's integral, which is defined as follows:

$$J_n(x) = \frac{1}{2\pi} \int_{-\pi}^{\pi} e^{i(x\sin\tau - n\tau)} \, d\tau. \tag{23}$$

Notably, Eq. equation 22 involves a term $\cos(\phi - \bar{\phi})$, in contrast to the $\sin \tau$ present in Bessel's integral. To reconcile this, we apply a change of variable $\tau = t - \frac{\pi}{2}$, which leads to:

$$
\begin{aligned}
J_n(x) &= \frac{1}{2\pi} \int_{-\pi}^{\pi} e^{i(x \sin \tau - n\tau)} \, d\tau \\
&= \frac{1}{2\pi} \int_{-\frac{1}{2}\pi}^{\frac{3}{2}\pi} e^{i(x \sin(t - \frac{\pi}{2}) - n(t - \frac{\pi}{2}))} \, d(t - \frac{\pi}{2}) \\
&= \frac{1}{2\pi} \int_{-\frac{1}{2}\pi}^{\frac{3}{2}\pi} e^{i(-x \cos t - nt)} e^{i\frac{\pi}{2}n} \, dt \\
&= \frac{i^n}{2\pi} \int_{0}^{2\pi} e^{-i(x \cos t + nt)} \, dt.
\end{aligned}
\tag{24}
$$

Setting $t = \phi - \bar{\phi}$ and substituting this modified integral into Eq. equation 22, we derive the following expression for the Fourier transform of the basis function:

$$
\begin{aligned}
\mathcal{F}\{p\}_{(\omega_\phi, \alpha_\rho + i\omega_\rho)}(\bar{\phi}, \bar{\rho}) &= \int_0^\infty \rho^{\alpha_\rho + i\omega_\rho} \int_0^{2\pi} e^{-i(2\pi\rho\bar{\rho}\cos(\phi - \bar{\phi}) - \omega_\phi \phi)} \, d\phi\rho \, d\rho \\
&= \int_0^\infty \rho^{\alpha_\rho + i\omega_\rho} \int_0^{2\pi} e^{-i(2\pi\rho\bar{\rho}\cos(\phi - \bar{\phi}) - \omega_\phi(\phi - \bar{\phi}) - \omega_\phi \bar{\phi})} d(\phi - \bar{\phi})\rho \, d\rho \\
&= \int_0^\infty \rho^{\alpha_\rho + i\omega_\rho} \int_0^{2\pi} e^{-i(2\pi\rho\bar{\rho}\cos t - \omega_\phi t - \omega_\phi \bar{\phi})} \, dt\rho \, d\rho \\
&= 2\pi (\frac{1}{i})^{-\omega_\phi} e^{i\omega_\phi \bar{\phi}} \int_0^\infty \rho^{\alpha_\rho + i\omega_\rho} J_{-\omega_\phi}(2\pi\rho\bar{\rho})\rho \, d\rho
\end{aligned}
\tag{25}
$$

By employing the property of Bessel's function, as stated in Eq. equation 26:

$$
J_{-n}(x) = (-1)^n J_n(x).
\tag{26}
$$

we can reformulate Eq. equation 25. Utilizing this property allows us to address the sign of the Bessel function's order in the integral. Thus, Eq. equation 25 is rewritten as:

$$
\mathcal{F}\{p\}_{(\omega_\phi, \alpha_\rho + i\omega_\rho)}(\bar{\phi}, \bar{\rho}) = 2\pi (-i)^{\omega_\phi} e^{i\omega_\phi \bar{\phi}} \int_0^\infty \rho^{\alpha_\rho + i\omega_\rho} J_{\omega_\phi}(2\pi\rho\bar{\rho})\rho \, d\rho
\tag{27}
$$

To compute Eq. equation 27, the Hankel transform is utilized, which is defined as:

$$
H_n(k) = \int_0^\infty f(\rho) J_n(k\rho)\rho \, d\rho.
\tag{28}
$$

For a function $f(\bar{\rho}) = \bar{\rho}^s$, where $\bar{\rho} \in \mathbb{R}^+$ and $s \in \mathbb{C}$, its Hankel transform **?** can be expressed as:

$$
H_n(k) = \mathcal{H}\{f\}(k) = \frac{2^{s+1}}{k^{s+2}} \frac{\Gamma\left(\frac{1}{2}(2 + n + s)\right)}{\Gamma\left(\frac{1}{2}(n - s)\right)},
\tag{29}
$$

where $\Gamma$ denotes the Gamma function and the real part of $s$ falls within the interval $(-2, -0.5)$. Substituting the integral over $\rho$ in Eq. equation 25 with this formulation of the Hankel transform, and setting $k = 2\pi\bar{\rho}$ and $s = \alpha_\rho + i\omega_\rho$, we derive:

$$
\mathcal{F}\{p\}_{(\omega_\phi, \alpha_\rho + i\omega_\rho)}(\bar{\phi}, \bar{\rho}) = \frac{\pi(-i)^{\omega_\phi} e^{j\omega_\phi \bar{\phi}}}{(\pi\bar{\rho})^{2 + \alpha_\rho + i\omega_\rho}} \frac{\Gamma\left(\frac{1}{2}(2 + \omega_\phi + \alpha_\rho + i\omega_\rho)\right)}{\Gamma\left(\frac{1}{2}(\omega_\phi - \alpha_\rho - i\omega_\rho)\right)}.
\tag{30}
$$

To circumvent the issue of negative integers in the Gamma functions within our calculations, we can apply the property outlined in Eq. equation 26. This property ensures that the arguments within the Gamma functions remain within their defined domain, thus maintaining the mathematical validity of the expressions. In addition, we represent the angular frequency in its general form as $a\omega_\phi$. Consequently, the Fourier transform, incorporating this generalized angular frequency, is given by:

$$
\mathcal{F}\{p\}_{(\omega_\phi, \alpha_\rho + i\omega_\rho)}(\bar{\phi}, \bar{\rho}) = \frac{\pi(-i)^{|\omega_\phi|} e^{ja\omega_\phi \bar{\phi}}}{(\pi\bar{\rho})^{2 + \alpha_\rho + i\omega_\rho}} \frac{\Gamma\left(\frac{1}{2}(2 + |\omega_\phi| + \alpha_\rho + i\omega_\rho)\right)}{\Gamma\left(\frac{1}{2}(|\omega_\phi| - \alpha_\rho - i\omega_\rho)\right)}.
\tag{31}
$$

## B  SIMILARITY GROUP CONVOLUTIONS

In this section, we utilize $T_{(\theta,r,a)}$ to denote transformations consisting of rotation by $\theta$ and scaling by $r$ about the origin at $(0,0)$, and horizontal reflection indicator $a$. Additionally, we only derive the discrete similarity group convolutions. The continuous translation, rotation and scaling subgroup convolution can be easily derived by setting the reflection indicator to be constant 1.

### B.1  DERIVATIONS OF SIMCONV THAT LIFTS FUNCTION SPACES

Let $(\boldsymbol{x},\theta,r,a) \in \mathbb{R}^2 \times S^1 \times \mathbb{R}^+ \times \{\pm 1\}$ be the coordinates, $f' : \mathbb{R}^2 \to \mathbb{C}$ be a 2D function and $g' : \mathbb{R}^2 \to \mathbb{C}$ be a 2D convolution kernel. The similarity group convolution lifts the representation space from the 2D image plane to the 5D space $\mathbb{R}^2 \times S^1 \times \mathbb{R}^+ \times \{\pm 1\}$ by convolving the input $f'$ with a set of rotated, scaled and flipped $g'$. The similarity transformation other than translation can be achieved by representing $g'$ as the linear combination of the basis functions.

$$g'(\boldsymbol{x}) = \sum_{\omega_\phi,\omega_\rho} G'(\omega_\phi,s_\rho)p_{(\omega_\phi,s_\rho)}(\boldsymbol{x}), \tag{32}$$

$$T_{(\theta,r,a)}\{g'\}(\boldsymbol{x}) = \sum_{\omega_\phi,\omega_\rho} G'(\omega_\phi,s_\rho)T_{(\theta,r,a)}\{p\}_{(\omega_\phi,s_\rho)}(\boldsymbol{x})$$
$$= \sum_{\omega_\phi,\omega_\rho} G'(\omega_\phi,s_\rho)p_{(\omega_\phi,s_\rho,a)}(\boldsymbol{x})e^{-i\omega_\phi\theta}r^{-s_\rho}, \tag{33}$$

where $s_\rho = \alpha_\rho + i\omega_\rho$, $G'(\omega_\phi,s_\rho) = \mathcal{M}\{g'\}(\omega_\phi,s_\rho)$ is the AFMT of the function $g'$ and $p_{(\omega_\phi,s_\rho)}(\phi,\rho,a) = e^{ia\omega_\phi\phi}\rho^{s_\rho}$ is the AFMT basis function. It follows that the simConv can be implemented as the linear combination of convolutions between the input $f'$ and the basis function $p$.

$$(f *_{\text{sim}(2)} g)(\boldsymbol{x},\theta,r,a) = \sum_{\omega_\phi,\omega_\rho} (f' * p_{(\omega_\phi,s_\rho)})(\boldsymbol{x},a)G'(\omega_\phi,s_\rho)e^{-i\omega_\phi\theta}r^{-s_\rho}. \tag{34}$$

Finally, the right-hand side of Eq. equation 34 is the inverse AFMT w.r.t. frequencies $(-\omega_\rho,-s_\rho)$. Moving it to the left-hand side yields the frequency domain representation.

$$\mathcal{M}\{f' \circledast g'\}_{(-\omega_\phi,-s_\rho)}(\boldsymbol{x},a) = \{f * p_{(\omega_\phi,s_\rho)}\}(\boldsymbol{x},a)G'(\omega_\phi,s_\rho). \tag{35}$$

### B.2  DERIVATIONS OF SIMCONV

Similar to the simConv that lifts function spaces, the similarity transformation other than translation can be achieved by representing $g$ as the linear combination of the basis functions in $\mathbb{R}^2 \times S^1 \times \mathbb{R}^+ \times \{\pm 1\}$.

$$T_{(\phi,\rho,a)}\{g\}(\boldsymbol{x},\theta,r,b) = \sum_{\omega_\phi,\omega_\theta,\omega_\rho,\omega_r} \begin{Bmatrix} G(\omega_\phi,s_\rho,\omega_\theta,s_r,ab) \\ p_{(\omega_\phi-\omega_\theta,\alpha_\rho-\alpha_r+i(\omega_\rho-\omega_r),a)}(\boldsymbol{x}) \\ p_{(\omega_\theta,\alpha_r+i\omega_r,a)}(\theta,r) \\ e^{-i\omega_\phi\phi}\rho^{-(\alpha_\rho+i\omega_\rho)} \end{Bmatrix} \tag{36}$$

where $s_\rho = \alpha_\rho + i\omega_\rho$, $s_r = \alpha_r + i\omega_r$ and $G(\omega_\phi,s_\rho,\omega_\theta,s_r,b) = \mathcal{M}\{g\}(\omega_\phi,\alpha_\rho+i\omega_\rho,\omega_\theta,\alpha_r+i\omega_r)$ is the AFMT coefficients w.r.t. both $\mathbb{R}^2$ and $S^1 \times \mathbb{R}^+$. Let $F_{(\omega_\phi,s_\rho)}(\boldsymbol{y},a)$ be the AFMT coefficients of the function $f$ w.r.t $S^1 \times \mathbb{R}^+$. Substitute Eq. equation 36 to Eq. 4 and represent the function $f$ as

the inverse AFMT of $F$,

$$
\left\{ f *_{\text{sim}(2)} g \right\}(\boldsymbol{x}, \theta, r, b)
$$

$$
= \sum_{\omega_\phi, \omega_\theta, \omega_\rho, \omega_r, a} \left\{ \begin{array}{l} G(\omega_\phi, s_\rho, \omega_\theta, s_r, ab) \\[4pt] p_{(\omega_\theta, \alpha_r + i\omega_r)}(\theta, r, a) \\[4pt] \displaystyle\int_{\mathbb{R}^2 \times S^1 \times \mathbb{R}^+} \left\{ \begin{array}{l} f(\boldsymbol{y}, \phi, \rho, a) e^{-i\omega_\phi \phi} \rho^{-(\alpha_\rho + i\omega_\rho)} \\[4pt] p_{(\omega_\phi - \omega_\theta, \alpha_\rho - \alpha_r + i(\omega_\rho - \omega_r))}(\boldsymbol{x} - \boldsymbol{y}, a) \end{array} \right\} d\boldsymbol{y} d\phi \, \dfrac{d\rho}{\rho} \end{array} \right\} d\boldsymbol{y}
$$

$$
= \sum_{\omega_\phi, \omega_\theta, \omega_\rho, \omega_r, a} \left\{ \begin{array}{l} G(\omega_\phi, s_\rho, \omega_\theta, s_r, ab) \\[4pt] p_{(\omega_\theta, \alpha_r + i\omega_r, a)}(\theta, r) \\[4pt] \left\{ F_{(\omega_\phi, s_\rho)}(a) * p_{(\omega_\phi - \omega_\theta, \alpha_\rho - \alpha_r + i(\omega_\rho - \omega_r))} \right\}(\boldsymbol{x}, a) \end{array} \right\}
$$

$$
= \mathcal{M}^{-1} \left\{ \sum_{\omega_\phi, \omega_\rho, a} \left\{ \begin{array}{l} G(\omega_\phi, s_\rho, \omega_\theta, s_r, ab) \\[4pt] \left\{ F_{(\omega_\phi, \omega_\rho)}(a) * p_{(\omega_\phi - \omega_\theta, \alpha_\rho - \alpha_r + i(\omega_\rho - \omega_r))} \right\}(\boldsymbol{x}, a) \end{array} \right\} \right\}
$$

$$
\tag{37}
$$

where $\mathcal{M}^{-1}$ computes the inverse AFMT using the basis function $p_{(\omega_\theta, \alpha_r + i\omega_r)}(\theta, r, a)$. Moving $\mathcal{M}^{-1}$ to the left-hand side of the equation yields

$$
\mathcal{M}\left\{ f *_{\text{sim}(2)} g \right\}(\boldsymbol{x}, \omega_\theta, \alpha_r + i\omega_r, b)
$$

$$
= \sum_{\omega_\phi, \omega_\rho, a} \left\{ \begin{array}{l} G(\omega_\phi, s_\rho, \omega_\theta, s_r, b) \\[4pt] \left\{ F_{(\omega_\phi, \omega_\rho)}(a) * p_{(\omega_\phi - \omega_\theta, \alpha_\rho - \alpha_r + i(\omega_\rho - \omega_r))} \right\}(\boldsymbol{x}, a) \end{array} \right\}
$$

$$
\tag{38}
$$

## C SECNN IMPLEMENTATION DETAILS

### C.1 GROUP POOLING

The orientation histogram plays a role in transforming equivariant feature maps into non-invariant representations. This is necessary for natural image classification, as invariant representation, especially orientation-invariant representation, is not always required. Previous works have adopted different approaches to achieve this. For example, the orientation-equivariant E2CNN Weiler & Cesa (2019) reduces the network's equivariance by gradually reducing the number of orientation samples.

### C.2 SECNN ARCHITECTURES

Table 4: Baseline and SECNN architectures used in this paper. Each cell represents the weight spaces and the number of output channels. The titles "3D," "4D," and and "Mix" indicate the type of weight space mixture used in the SECNNs (see Section 4 on Trading Off Full Equivariance for More Features and Hence Accuracy). Brackets containing two convolution layers represent a ResNet block.

| | MNIST | | | CIFAR | |
|---|---|---|---|---|---|
| Baseline | 4D | 3D | Mix | WRN16-8 | Mix |
| 24 | $\mathbb{R}^2, 8$ | $\mathbb{R}^2, 18$ | $\mathbb{R}^2, 17$ | 16 | $\mathbb{R}^2, 16$ |
| 32 max-pool | $\mathbb{R}^2 S^1 \mathbb{R}^+, 8$ max-pool | $\mathbb{R}^2 S^1, 18$ max-pool | $\mathbb{R}^2 S^1 \mathbb{R}^+, 17$ scale max-pool max-pool | $\begin{bmatrix} 128 \\ 128 \end{bmatrix} \times 2$ | $\begin{bmatrix} \mathbb{R}^2 S^1 \mathbb{R}^+\{\pm 1\}, 16 \\ \mathbb{R}^2 S^1 \mathbb{R}^+\{\pm 1\}, 16 \end{bmatrix} \times 1$ scale max-pool $\begin{bmatrix} \mathbb{R}^2 S^1\{\pm 1\}, 16 \\ \mathbb{R}^2 S^1\{\pm 1\}, 16 \end{bmatrix} \times 1$ |
| 36 36 max-pool | $\mathbb{R}^2 S^1 \mathbb{R}^+, 16$ $\mathbb{R}^2 S^1 \mathbb{R}^+, 16$ max-pool | $\mathbb{R}^2 S^1, 36$ $\mathbb{R}^2 S^1, 36$ max-pool | $\mathbb{R}^2 S^1, 34$ $\mathbb{R}^2 S^1, 34$ max-pool | $\begin{bmatrix} 256 \\ 256 \end{bmatrix} \times 2$ | $\begin{bmatrix} \mathbb{R}^2 S^1\{\pm 1\}, 32 \\ \mathbb{R}^2 S^1\{\pm 1\}, 32 \end{bmatrix} \times 2$ |
| 64 96 | $\mathbb{R}^2 S^1 \mathbb{R}^+, 32$ $\mathbb{R}^2 S^1 \mathbb{R}^+, 32$ | $\mathbb{R}^2 S^1, 72$ $\mathbb{R}^2 S^1, 72$ | $\mathbb{R}^2 S^1, 68$ $\mathbb{R}^2 S^1, 68$ | $\begin{bmatrix} 512 \\ 512 \end{bmatrix} \times 2$ | $\begin{bmatrix} \mathbb{R}^2 S^1\{\pm 1\}, 64 \\ \mathbb{R}^2 S^1\{\pm 1\}, 64 \end{bmatrix} \times 2$ |
| | | | (group) pooling layer fully-connected layer | | |

## D    Experiment Settings and Runtime

### D.1    Translated, Rotated and Scaled MNIST

In rotation and scale equivariant CNN studies, the rotated MNIST (LeCun et al. (1998)) and scaled MNIST are frequently used datasets. We augmented the MNIST to created various combination of translated, rotated and scaled MNIST. Each dataset contains 10k training images, 2k for evaluation, and 50k for testing. Our results stem from training on a combined set of 12k images, pulling from both the training and evaluation datasets. To create the dataset, we augmented the MNIST-12k dataset by rotating and/or scaling images within range $[0, 2\pi]$ and $[0.3, 1]$ respectively. Additionally, we apply non-integer translation on the images along both $x-$ and $y-$axis within range $[-5, 5]$. All transformations are performed using bilinear interpolation to simulate continuous transformations. Consequently, we have several distinct MNIST datasets that are transformed by mixtures of these transformations.

The augmentation for the rotated MNIST involves random rotation within the range of $[0, 2\pi]$. For the scaled MNIST, we applied random scaling within the range of $[0.5, 2]$. Depend on the mixture of transformations, we combined the two types of augmentations. We did not perform training-time augmentation in translation as the PyTorch's RandomAffine function implements discrete translation. The images are also mean and standard deviation normalized.

### D.2    CIFAR10/100

In our experiments involving the natural image datasets CIFAR10/100 (Krizhevsky et al. (2009)), each dataset contains 50,000 training images and 10,000 test images, distributed across 10 and 100 categories, respectively. We evaluated both the proposed networks and the baseline models with specific training augmentations. These augmentations encompassed random translation, random horizontal flipping, as well as mean and standard deviation normalization.

### D.3    SECNNs and Training Parameters

The SECNNs use $5 \times 5$ cropped shiftable basis, which are computed from $1024 \times 1024$ spatial frequencies ($\omega_{\boldsymbol{x}}$), to implement simConv. For images with input size $n \times n$, the spatial period $X = 2n$ and scale period $S = \sqrt{2}n$. The number of orientation and scale frequencies ($\omega_\phi$, $\omega_\theta$, $\omega_\rho$ and $\omega_r$) is 5. There are 16 orientations (8 for CIFAR experiments) uniformly sampled in $[0, 2\pi)$ and 8 scales uniformly sampled in $[0.5, 8]$. Finally, $\alpha_\rho = -1$ and $\alpha_r = 0$.

All models are trained for 200 epochs with a batch size of 128. We use SGD optimizer with Nesterov momentum of 0.9 and weight decay of 0.0005. The initial learning rate is set to 0.1 and divided by 5 after 60, 120 and 180 epochs.

### D.4    SECNNs Runtime

Table 5: Runtime on CIFAR10/100, Baselines: 1×A100, SECNNs: 8×A100

|  | WRN | SESN | E2CNN | SECNNs |
|---|---|---|---|---|
| Speed (seconds / epoch) | 23 | 93 | 128 | 355 |
| GPU memory (GB) | 8 | 8 | 11 | 328 |

