# OpenReview forum: "Similarity Group Equivariant Convolutional Networks"
_ICLR.cc/2025/Conference — Submitted to ICLR 2025_

### Official Review · Reviewer_TFxn · 2024-11-03

**Soundness:** 3
**Presentation:** 3
**Contribution:** 2
**Rating:** 5
**Confidence:** 4

**Summary:**

This paper constructs similarity group equivariant convolutional networks by utilizing steerable, (approximately) shiftable and scalable kernels. The authors demonstrate how to improve computational efficiency and present some design choices. They implement SECNNs that are equivariant to a continuous subgroup and a discrete version of the similarity group, and evaluate them on transformed MNIST and CIFAR10/100, with results demonstrating improved empirical performance.

**Strengths:**

1. The paper constructs equivariant networks, called SECNNs, for the full similarity group using shiftable, steerable, and scalable filters. This achieves equivariance with respect to continuous translation (often ignored in prior work), rotation, scale, and reflection.
2. The paper improves the computational efficiency of SECNNs with multidimensional kernels through measures including a cropped Fourier series of the steerable and scalable basis.
3. The paper provides comprehensive details on the model and experimental setup, ensuring a fair evaluation of the proposed approach.

**Weaknesses:**

1. While SECNN extends Zhang & Williams (2022) to include reflection, this extension alone may not constitute a substantial advancement for the field. A more detailed and explicit discussion on the limitations of Zhang & Williams (2022), the specific advantages of SECNN, and the technical challenges involved in achieving these improvements would enhance the paper.
2. Existing methods achieving affine or general Lie group equivariance [1-3] are not adequately compared, leaving a gap in understanding how SECNN competes with these models. Similarity equivariance might also be achievable through these alternatives.
3. One of the paper's key contributions is achieving equivariance for the full similarity group, including reflection. However, the implementation and experiments do not cover the full similarity group with both reflection and continuous rotation.
4. RST-CNN and SREN, which offer rotation, translation, and scale equivariance, would serve as more relevant baselines than the chosen ones (SESN and E2CNN). Including comparisons with RST-CNN or/and SREN could be more convincing.


References

[1] Lachlan E MacDonald, Sameera Ramasinghe, and Simon Lucey. Enabling equivariance for arbitrary Lie groups. CVPR 2022.

[2] Mircea Mironenco and Patrick Forre ́. Lie group decompositions for equivariant neural networks. ICLR 2024.

[3] Yikang Li, Yeqing Qiu, Yuxuan Chen, Lingshen He, and Zhouchen Lin. Affine equivariant networks based on differential invariants. CVPR 2024.

**Questions:**

1. It could be important to provide a comprehensive discussion on the limitations of Zhang & Williams (2022) and improvements of this paper.
2. A comparison with existing methods that achieve affine or general Lie group equivariance is necessary (see Weakness 2). As these methods can also attain similarity group equivariance, what distinct advantages does SECNN offer?
3. Experiments on the full similarity group would strengthen the paper. Minimally, it should encompass a brief discussion on the practicality of implementing the full similarity group, along with rationale and insights on selecting different groups to maintain equivariance for different tasks.
4. Comparisons with RST-CNN and/or SREN in the experimental section would offer a more convincing evaluation than the current baselines.
5. Could you further explain how making filters “shiftable, steerable and scalable” in this paper contributes to the achievement of equivariance?
6. Providing examples of promising applications for similarity group equivariant convolutional networks would be both valuable and intriguing.
7. On line 358, 2D images are modeled as functions $\mathbb{Z}^2 \rightarrow \mathbb{R}$. But defining similarity transformations on functions defined over $\mathbb{Z}^2$ is impractical, which hinders the discussion of similarity equivariance. It may be more reasonable to regard 2D images as functions $\mathbb{R}^2 \rightarrow \mathbb{R}$.
8. For natural image classification, the paper employs an orientation histogram approach to convert equivariant feature maps into non-invariant representations. In this context, what role does the equivariance of features play?
9. In Table 2, it is somewhat unexpected that SESN performs worse on Scaled MNIST compared to CNN and E2CNN. Is there an explanation for this?

---

> ### Author Response · Authors · 2024-11-24
>
> Thank you for the insightful review. We are pleased to answer your questions and will revise the manuscript based on your feedback.
>
> > 1. It could be important to provide a comprehensive discussion on the limitations of Zhang & Williams (2022) and improvements of this paper.
> We appreciate the reviewer's suggestion to provide a comprehensive discussion on the limitations of Zhang & Williams (2022) and how our work improves upon it.
>
> **Limitations of Zhang & Williams (2022):**
> 1. **Limited Experimental Scope and Shallow Implementation:**
> Zhang & Williams (2022) conducted experiments involving only a single convolution with a precomputed kernel. This shallow implementation does not fully explore the potential of similarity group convolutions in deep neural networks.
>
> 2. **Inefficient Frequency Domain Implementation:**
> Their implementation of group convolution exclusively in the frequency domain proves to be computationally inefficient and limits performance. For example, processing a $32\times 32$ CIFAR image requires handling all $32\times 32$ frequency components (or more), resulting in significantly higher GPU memory consumption compared to our proposed spatial domain implementation, which uses cropped $5\times 5$ Fourier series.
>
> 3. **Lack of Scalability Due to Non-Periodic Kernels:**
> The kernels are not assumed to be periodic with respect to the logarithmic scale, which hampers scalability. Without periodicity, continuous scaling requires extensive sampling of scale frequencies (real numbers), making it impractical for efficient computation.
>
> **Our Contributions and Improvements:**
>
> 1. **Deep Similarity Group Convolution Networks:**
> We constructed deep neural networks utilizing similarity group convolutions and tested them on popular benchmark datasets. This demonstrates the practical applicability and effectiveness of our approach in real-world scenarios.
>
> 2. **Efficient Spatial Domain Implementation:**
> While implementing group convolutions in the frequency domain is theoretically elegant, we observed that it is computationally inefficient and yields suboptimal performance (as discussed in Lines 396–405 of our submission). By using cropped $5\times 5$ Fourier series as kernels in the spatial domain, we achieve computational efficiency without compromising performance. This approach significantly reduces GPU memory usage and computational overhead.
>
> 3. **Enhanced Translation Invariance:**
> Previous research, such as Azulay & Weiss (2019), has shown that downsampling and max-pooling can disrupt translation invariance. Other studies (e.g., [4]) have used adaptive max-pooling to preserve it. Our method achieves translation invariance theoretically and enhances it in practice through the use of 5×5 kernels, downsampling feature maps, and conventional max-pooling.
>
> 4. **Scalable Periodic Kernels in Log-Scale:**
> By assuming that the log-scale of kernels is periodic, we enable scalability by manipulating continuous scaling with a finite set of discrete representations. Although periodicity in scale may not have a direct physical interpretation, it is practically useful. This assumption eliminates the need to sample scale frequencies extensively, thus enhancing computational efficiency.
>
> **Summary:**
>
> While Zhang & Williams (2022) introduced the concept of similarity group convolution using AFMT basis in an arXiv preprint (not peer-reviewed), our work significantly advances this idea by:
> * Developing deep similarity group convolutional networks.
> * Implementing an efficient spatial domain approach with cropped Fourier series kernels.
> * Demonstrating enhanced translation invariance and scalability.
> * Validating our methods on popular benchmark datasets, showcasing practical improvements over prior work.
>
> We believe these contributions address the limitations of Zhang & Williams (2022) and represent a meaningful advancement in the field.
>
> [4] Zhang, Richard. "Making convolutional networks shift-invariant again." International conference on machine learning. PMLR, 2019.

---

> ### Author Response · Authors · 2024-11-24
>
> > 2. A comparison with existing methods that achieve affine or general Lie group equivariance is necessary (see Weakness 2). As these methods can also attain similarity group equivariance, what distinct advantages does SECNN offer?
>
> 1. **Lie Group and Lie Algebra Approache ([1], [2]):**
>
> These methods employ Lie group theory and Lie algebras to achieve equivariance by learning convolution kernels through MLPs and performing Monte Carlo integration over the group transformations.
>
> * **Limitations:**
> * *MLPs are function approximator:* The kernel learned by MLP may not achieve exact equivariance.
> * *Stochastic Integration:* Monte Carlo integration is inherently stochastic and may require a large number of samples to approximate the integrals accurately, leading to increased computational cost and potential variance in performance.
> * *Explainability:* The use of MLPs for kernel learning introduces complexity and limits interpretability. The learned kernels are not easily explainable in terms of established signal processing theories.
>
> 2. **Differential Invariants Approach ([3]):**
>
> This approach constructs invariant representations using differential invariants, effectively discarding transformation-related variations.
> * **Limitations:**
>
> * *Information Loss:* Invariance is a special case of equivariance. However, invariant representations inherently discard information about the transformations (e.g., scale and orientation), which can limit the expressiveness and flexibility of the model.
>
> **Advantages of SECNN:**
> 1. **Deterministic Integration with Fourier Analysis:**
> Our method employs Fourier analysis and deterministic integration, grounded in well-established theories like the Fourier Transform and Nyquist–Shannon Sampling Theorem. We interpret similarity group equivariance in terms of signal bandwidth and joint localization (see Lines 304–308), providing clear insights into the model's behavior.
> 2. **Bandlimited Kernels Using Finite Basis Functions:**
> By using a finite number (5×5) of basis functions, SECNN ensures convolution kernels are bandlimited with respect to orientation and log-scale.
> 3. **Enhanced Explainability and Balanced Theory and Practical Performance:**
> While bandlimited kernels may not capture all high-frequency details in images, they provide a balance between theoretical rigor and practical effectiveness. Our experimental results demonstrate that this trade-off leads to strong performance on benchmark datasets.
>
> > 3. Experiments on the full similarity group would strengthen the paper ...
>
> Based on our response to Question 1, we believe our experimental design is sufficient to demonstrate the contributions of this paper. Moreover, we followed the experimental designs of the E2CNN and SESN papers and used their conclusions to guide group selection for different tasks. For MNIST classification, the E2CNN paper shows that translation-rotation equivariance is better than translation-rotation-reflection equivariance, while on CIFAR10/100, it shows the reverse result. Therefore, we chose to implement translation-rotation-scaling group equivariance in our MNIST experiments and similarity group equivariance in our CIFAR10/100 experiments.
>
> >4. Comparisons with RST-CNN and/or SREN in the experimental section ...
>
> We could not find the source code for SREN. The MNIST experiment in the SREN paper focuses on out-of-distribution performance—that is, training on the original MNIST without data augmentation and then testing on transformed MNIST. They also show performance improvement when using data augmentation during training. However, their experimental settings—such as network architecture, image size, and transformation parameters—are different from those in the SESN and E2CNN experiments. To compare SECNNs with SREN, we would have to conduct entirely different experiments. We decided not to do this for two reasons:
> 1. The ResNet-18 architecture is overkill for the MNIST dataset.
> 2. As they also show that data augmentation can improve SREN's performance, out-of-distribution performance is not very important in terms of similarity transformations. This is because the computational overhead of data augmentation with similarity transformations is negligible. To maximize performance, SREN needs data augmentation. In fact, it is common in previous works to use data augmentation to improve the performance of group CNNs. Group CNNs need data augmentation because, when augmenting datasets like MNIST using bilinear interpolation, the transformations are not aliasing-free.
> The code of RST-CNN is based on that of SESN. Although RST-CNN uses a different basis function, the discrete scale convolution part is the same as in SESN. It is very likely that RST-CNN will suffer from the same scale instability issue, as discussed in our answer to Question 9.
> Both E2CNN and SESN use the same approach as SECNN. For these reasons, E2CNN and SESN are the best baselines.

---

> ### Author Response · Authors · 2024-11-24
>
> >5. Could you further explain how making filters “shiftable, steerable and scalable” in this paper contributes to the achievement of equivariance?
>
> Equivariance is achieved by implementing the similarity group action defined in Equation (3) and utilizing the integral defined in Equation (4). By making the filter $g$ in Equation (4) "shiftable, steerable, and scalable," we can continuously transform the filter to handle continuous translations, rotations, and scalings in real-world applications while using finite and discrete representations.
>
> > 6. Providing examples of promising applications for similarity group equivariant convolutional networks would be both valuable and intriguing.
>
> Feature matching is the cornerstone of the structure-from-motion pipeline, which reconstructs a 3D model from a set of 2D images with unknown camera poses. Recent feature matching methods use conventional CNNs to train a Siamese network on pairs consisting of an image and its transformed version. The transformations used for data augmentation involve translation, rotation, and uniform scaling [5–7]. Therefore, equivariance to the similarity group (without reflection) is a desirable property in this application.
>
> >7. On line 358, 2D images are modeled as functions Z2→R. But defining similarity transformations on functions defined over Z2 is impractical, which hinders the discussion of similarity equivariance. It may be more reasonable to regard 2D images as functions R2→R.
>
> We agree with your assessment. We are dealing with discrete samples of a continuous signal. Therefore, representing it as a function from R2$\to$R is the most accurate description.
>
> > 8. For natural image classification, the paper employs an orientation histogram approach to convert equivariant feature maps into non-invariant representations. In this context, what role does the equivariance of features play?
>
> The role of equivariance in features is to reduce the number of parameters needed or to increase parameter efficiency. This is why previous works, such as SESN and E2CNN, maintain a similar number of parameters in their network experiments. When LeCun et al. (1998) proposed CNNs, they stated that CNNs needed fewer parameters because of weight sharing. The same is true for orientation and scale. CNNs use weight sharing with respect to position, while group CNNs use weight sharing with respect to position, orientation, and scale. Why should translation be more special than rotation and scaling? I believe it should not be, even though images are uniformly sampled on a rectangular grid.
>
> Note that the orientation histogram is computed only after the last convolutional layer. The non-invariant histogram representations are mainly inspired by the SIFT descriptor, which is still being used in the most popular structure-from-motion software, Colmap, due to its efficiency and performance. The proposed SECNNs can learn shiftable, steerable, and scalable kernels to produce orientation histograms similar to SIFT. I think that SECNNs and orientation histograms have potential and can achieve more than what SIFT can do.
>
> > 9. In Table 2, it is somewhat unexpected that SESN performs worse on Scaled MNIST compared to CNN and E2CNN. Is there an explanation for this?
>
> In the SESN paper, the MNIST experiment uses a three-layer convolutional architecture. It reports two SESN results: SESN-Scalar and SESN-Vector. The SESN-Vector, which utilizes vector features, is intended to preserve more scale information than SESN-Scalar, and the results have confirmed this. However, in the STL10 experiment, SESN-A (which uses vector features) performs worse than SESN-B (which uses scalar features).
>
> In our MNIST experiment, we use a seven-layer architecture with SESN-Vector. We suspect that there is some instability in the vector features, and the seven-layer architecture amplifies this effect. Similarly, our SECNN-4D, which uses scale vector representations, performs worse than SECNN-3D, which pools the scale vector into a scalar. However, SECNN-4D still outperforms the CNN baseline.
>
> [5] DeTone, Daniel, Tomasz Malisiewicz, and Andrew Rabinovich. "Superpoint: Self-supervised interest point detection and description." Proceedings of the IEEE conference on computer vision and pattern recognition workshops. 2018.
>
> [6] Sarlin, Paul-Edouard, et al. "Superglue: Learning feature matching with graph neural networks." Proceedings of the IEEE/CVF conference on computer vision and pattern recognition. 2020.
>
> [7] Gleize, Pierre, Weiyao Wang, and Matt Feiszli. "Silk: Simple learned keypoints." Proceedings of the IEEE/CVF international conference on computer vision. 2023.

---

> ### Comment · Reviewer_TFxn · 2024-11-28
>
> Thank you for the response. I have a few follow-up points I would like to discuss.
>
> 1. Lie Group and Lie Algebra Approach. Based on my understanding, theoretically, equivariance in [1,2] does not depend on the MLP used to parameterize the kernel, and the equivariance error experiments in [1] appear to be reasonably good. Moreover, although sampling is required, [2] presents an improvement specifically targeting the number of samples. Do you have any additional insights on comparisons with these works?
>
> 2. Differential Invariants Approach. As mentioned in your response to Q9, vector representations are not necessarily superior to scalar ones. The MNIST experiments in SESN show only marginal differences. Thus, the conclusion regarding expressiveness might not be so straightforward, whereas flexibility could be a clearer benefit.
>
> 3. Given the identified limitations of [1-3], could you discuss scenarios or conditions where SECNN might perform better than the methods proposed in [1-3]? Any insights on specific tasks or setups would be valuable.
>
> 4. Regarding Q3, I am interested in understanding the circumstances in which discrete groups should be preferred over continuous groups, or vice versa. Could you share your perspective?
>
> 5. As an open-ended extension question, which techniques or methodologies proposed in this paper could possibly be adapted to other existing equivariant networks?
>
> 6. After reading your discussion with Reviewer uMhw, I also share some of the concerns, such as the clarification of the term "steerable." As suggested by uMhw, updating the paper with some of the clarifications would be ideal. If a full revision is not feasible at this stage, could you outline your plan for modifications or clarifications?
>
> [1] Lachlan E MacDonald, Sameera Ramasinghe, and Simon Lucey. Enabling equivariance for arbitrary Lie groups. CVPR 2022.
>
> [2] Mircea Mironenco and Patrick Forre ́. Lie group decompositions for equivariant neural networks. ICLR 2024.
>
> [3] Yikang Li, Yeqing Qiu, Yuxuan Chen, Lingshen He, and Zhouchen Lin. Affine equivariant networks based on differential invariants. CVPR 2024.

---

> ### Author Response · Authors · 2024-12-02
> **Modification Summary of Revised Submission for Reviewer TFxn**
>
> We appreciate your insightful questions, which have helped us better position our work. Thanks to the valuable reviews from all reviewers, we have updated and uploaded the manuscript.
>
> * **Introduction:** We added a paragraph introducing the Lie group and Lie algebra approach and discussed its limitations in achieving continuous translation equivariance. Currently, methods in both approaches focus on continuous equivariance with respect to the group $H$ of {$R^n, +$} $\rtimes H$. Specifically, these methods focus on continuous transformations $A^{-1} x$ in the $R^2$ plane (Eq. (3) of our manuscript). This is primarily because the translation group {$R^n, +$} is a non-compact group. Our approach extends the scope of continuous transformations from $A^{-1}x$ to $A^{-1} x – y$. Additionally, we refined the contribution list, highlighting our contributions relative to Zhang & Williams (2022), as well as the application scenarios, such as computer vision tasks involving 2D images.
>
> * **Related Works:** We have enhanced the discussion of the Lie group and Lie algebra approach by including more recent works. Furthermore, we explicitly compare our method’s contribution relative to these two approaches. Compared to methods in the steerable CNNs approach, we derive a Fourier basis for the similarity group and demonstrate the effectiveness of this basis by applying a deep SECNN to image classification. Compared to methods in both the steerable CNNs and Lie group Lie algebra approaches, our method achieves continuous translation equivariance because we derive the Fourier transform of the basis functions defined in the log-polar coordinate system, which allows us to construct a Fourier series for these functions in the $R^2$ plane. Lastly, we have added more works to Table 1 for a more comprehensive comparison.
>
> * **Translated, Rotated and Scaled MNIST Experiment:** We have added the baseline sim2CNN [Knigge et al., 2022] to this experiment. For the sake of brevity, we report results from only one run for each dataset due to the revision deadline. Additionally, we did not match the number of parameters of sim2CNN with the others. This decision was made because [Knigge et al., 2022] claimed state-of-the-art results on rotated MNIST, and we used this as a baseline to show that even a small amount of (non-integer) translation can significantly degrade performance.
> * **Background: Shiftable, Steerable and Scalable Filters** We have clarified that shiftable, steerable, and scalable filters are, in fact, steerable filters with respect to translation, rotation, and scaling.
>
> > 1. Lie Group and Lie Algebra Approach. Based on my understanding, theoretically, equivariance in [1,2] does not depend on the MLP used to parameterize the kernel, and the equivariance error experiments in [1] appear to be reasonably good. Moreover, although sampling is required, [2] presents an improvement specifically targeting the number of samples. Do you have any additional insights on comparisons with these works?
>
> In theory, MLPs are function approximators, where different configurations of layers and channels can lead to varying outputs or introduce errors. This is also how Finzi et al. 2020 compare their MLP-based approach with the steerable CNNs approach. However, in practice, all methods require some degree of approximation. The Lie group and Lie algebra approach offers greater flexibility in extending to other transformations, while the Fourier approach necessitates substantial effort to derive analytical basis functions on groups.
>
> That said, we advance the steerable CNNs approach by deriving the Fourier transform of the AFMT basis to incorporate continuous translation equivariance into SECNNs. In our MNIST experiment, we demonstrate that both E2CNN (from the steerable CNNs approach) and sim2CNN (from the Lie group and Lie algebra approach) experience a significant decrease in performance when dealing with translation and rotation transformations.
>
> > 2. Differential Invariants Approach ...
>
> We agree that flexibility is crucial. However, as noted in [3], its approach is limited to achieving only invariance. In the invariant MNIST variant classification experiment—a typical benchmark—methods from prior works in this field generally convert the feature map of the final convolutional layer into invariant representations. This approach may be sufficient in this specific case. However, in other computer vision tasks, such as feature matching, it may be necessary to extract feature descriptors from the last few convolutional layers. In such scenarios, vector representations— which can preserve more information than scalar representations—are often required. Moreover, SIFT, a vector representation of orientation, remains widely used in popular structure-from-motion software. Investigating this topic further is, however, beyond the scope of this paper.

---

> ### Author Response · Authors · 2024-12-02
> **Answers to Reviewer TFxn's Questions**
>
> > 3. Given the identified limitations of [1-3], could you discuss scenarios or conditions where SECNN might perform better than the methods proposed in [1-3]? Any insights on specific tasks or setups would be valuable.
>
> We have demonstrated that both E2CNN and sim2CNN [Knigge et al., 2022] experience performance degradation when both translation and rotation are involved. Since sim2CNN shares similarities with the approaches in [1,2], we believe that the methods in these works likely suffer from the same limitation. In tasks such as feature matching, where vector representations are essential and continuous translation-rotation-scale equivariance is required, we anticipate that SECNNs could outperform the methods presented in [1-3].
>
> > 4. Regarding Q3, I am interested in understanding the circumstances in which discrete groups should be preferred over continuous groups, or vice versa. Could you share your perspective?
>
> The objective was to achieve optimal performance on both MNIST and CIFAR10/100. In the case of MNIST, reflection invariance decreases accuracy, while in CIFAR10/100, it enhances it. Since reflection is a discrete transformation, the translation-reflection group (or Dihedral group) is a discrete group, and there is no continuous Dihedral group. Therefore, it is not the continuity of group equivariance that impacts performance, but rather the effect of reflection equivariance itself. In the MNIST case, Weiler and Cesa (2019) note that Dihedral invariance can introduce confusion between the digits 4 and 7 (as mentioned in footnote 11 on page 12). Conversely, in the CIFAR10/100 case, reflection invariance has been shown to improve accuracy, as reflection augmentation is commonly used as a standard data augmentation technique for conventional CNNs on these datasets.
>
> > 4. As an open-ended extension question, which techniques or methodologies proposed in this paper could possibly be adapted to other existing equivariant networks?
>
> One of our key contributions is demonstrating that the combination of translation and rotation can degrade the performance of existing equivariant networks. While deriving shiftable basis functions with respect to $R^n$  is a challenging task, we hope our approach will inspire future methods that incorporate continuous equivariance into networks, as well as show flexibility when dealing with non-uniformly sampled point cloud data. In the background section, we emphasize that basis functions must be periodic in $R^n$  and band-limited. We show that this can be approximated in practice, as the decay function within the radial profile of proposed basis functions behaves like $\rho^{-1}$, which is not ideally but effectively band-limiting in both the spatial and frequency domains.
>
> While WeilerCesa and Cesa 2019 chose a Gaussian radial profile, which yields approximately band-limited functions, the E2CNN still suffers from the translation-rotation equivariance issue. Thus, assuming periodicity in $R^n$  and ensuring that filters are bounded and approximately band-limiting is a necessary condition. We believe that methods employing MLPs to learn convolution kernels could also integrate this idea to improve their performance.
>
> David M Knigge, David W Romero, and Erik J Bekkers. Exploiting redundancy: Separable group convolutional networks on lie groups. In International Conference on Machine Learning, pp. 11359–11386. PMLR, 2022.

---

### Official Review · Reviewer_WwDC · 2024-11-04

**Soundness:** 3
**Presentation:** 2
**Contribution:** 2
**Rating:** 5
**Confidence:** 3

**Summary:**

This work presents SEC-NN, Similarity group equivariant convolutional networks for continuous translation, rotation, and scale equivariance. This is achieved by using an approximate translation and scale basis in the Joint-orientation Scale Space proposed by Zhang et al. The empirical experiments are conducted on MNIST, CIFAR10/CIFAR100 datasets and the work contains several design choices for the proposed method.

**Strengths:**

1. The paper presents a relevant related work section, motivating the area of research as well as building upon existing works.
2. The ablations provided for SECNNs in the paper and appendix is very thorough.

**Weaknesses:**

1. The writing of the paper can be improved as it lacks clarity on the work actually done creating difficulty in distinguishing it from existing work, especially in building simConv using shiftable steerable and scalable filters.
2. Existing methods is literature like Group convolution methods like [1] have steerable kernels, [2] present convolution kernels through weight sharing in position orientation space missing in the experiments comparison.
3.  [3] presents similar group equivariant convolutions using AFMT in join orientation-scale space. Given this, as far as I understand the proposed method, SECNNs use the same basis as [3] and provide a different implementation for similarity group convolutions and ablations for the same through different modeling choices.



[1] Implicit Convolutional Kernels for Steerable CNNs, Zhdanov et al
[2]Fast, Expressive Equivariant Networks through Weight-Sharing in Position-Orientation Space, Bekkers et al
[3] Similarity Equivariant Linear Transformation of Joint Orientation-Scale Space Representations, Zhang et al

**Questions:**

1. Could you clarify their contribution especially distinguishing from the work of Zhang et al. ?
2. Does the Fourier transform approach of the AFMT basis functions scale well? Are there other advantages of working in this representation scheme? The experiments presented are on MNIST and CIFAR10/CIFAR100 datasets, but would this approach be more useful in non-image domains?

---

> ### Author Response · Authors · 2024-11-25
> **Novelty**
>
> Thank you for the insightful review. We are pleased to answer your questions and will revise the manuscript based on your feedback.
>
> > **Weakness** 2. Existing methods is literature like Group convolution methods like [1] have steerable kernels, [2] present convolution kernels through weight sharing in position orientation space missing in the experiments comparison.
>
> The concept of steerable filter was first proposed in Freeman et al. (1991).  As discussed at line 190-204, Simoncelli and Freeman (Simoncelli et al. (1992)) generalized this concept to shiftable and scalable filters (see line 190-204). After the group equivariant CNN was introduced in Cohen & Welling (2016), the same authors combined the idea of steerable filter and group equivariance CNN, proposing steerable CNNs in 2017 [4].
>
> [1] combined the idea of steerable CNNs and implicit neural kernels, which basically are kernels learned by MLPs. The authors claimed that the proposed networks can achieve equivariance w.r.t. translations and any compact group. However, the scale group is not a compact group but a locally compact group (we have a related discussion at line 172-180). This means that their method cannot achieve scale equivariance and they have never claimed anything about scale equivariance.
>
> The same is true for [2], in which the title has explicitly stated “SE(n) Equivariant Networks”. The SE(n) group does not include the scale group. In your review, the group name “SE(n)” in the title is missing. Moreover, this work proposes the idea of an invariant attribute $a_{ij}$, which is associated with two points $(x_i, x_j)$. This idea works when group transformations are distance-preserving. Specifically, their focus group is $\text{SE}(n)$. When it comes to scaling, which is not a distance-preserving transformation, the invariant attributes given in the paper are not valid. For example, they use Euclidean distance as the invariant attribute for translation; however, the Euclidean distance is not invariant under scaling.
>
> In our approach, the scale group is locally compact, meaning that a Haar measure is unique up to a constant multiplier. It follows that we can deal with the scale factor by performing the integral $\int \frac{dr}{r}$ (see lines 173–176), achieving scale equivariance.
>
> Both [1] and [2] use MLPs to learn convolution kernels. This method is an approximation and does not achieve exact equivariance. Our method use a Fourier basis, which is a complete basis, to learn kernels. It can achieve exact equivariance in theory.
>
>
> [4] Cohen, Taco S., and Max Welling. "Steerable CNNs." International Conference on Learning Representations. 2017.

---

> ### Author Response · Authors · 2024-11-25
>
> > 1. Could you clarify their contribution especially distinguishing from the work of Zhang et al. ?
>
> We apologize for any confusion caused by the way we presented our work.  Zhang et al. (2022) is an arXin preprint but not a peer-reviewed paper.
>
> **Limitations of Zhang & Williams (2022):**
> 1. **Limited Experimental Scope and Shallow Implementation:**
> Zhang & Williams (2022) conducted experiments involving only a single convolution with a precomputed kernel. This shallow implementation does not fully explore the potential of similarity group convolutions in deep neural networks.
>
> 2. **Inefficient Frequency Domain Implementation:**
> Their implementation of group convolution exclusively in the frequency domain proves to be computationally inefficient and limits performance. For example, processing a $32\times 32$ CIFAR image requires handling all $32\times 32$ frequency components (or more), resulting in significantly higher GPU memory consumption compared to our proposed spatial domain implementation, which uses cropped $5\times 5$ Fourier series.
>
> 3. **Lack of Scalability Due to Non-Periodic Kernels:**
> The kernels are not assumed to be periodic with respect to the logarithmic scale, which hampers scalability. Without periodicity, continuous scaling requires extensive sampling of scale frequencies (real numbers), making it impractical for efficient computation.
>
> **Our Contributions and Improvements:**
>
> 1. **Deep Similarity Group Convolution Networks:**
> We constructed deep neural networks utilizing similarity group convolutions and tested them on popular benchmark datasets. This demonstrates the practical applicability and effectiveness of our approach in real-world scenarios.
>
> 2. **Efficient Spatial Domain Implementation:**
> While implementing group convolutions in the frequency domain is theoretically elegant, we observed that it is computationally inefficient and yields suboptimal performance (as discussed in Lines 396–405 of our submission). By using cropped $5\times 5$ Fourier series as kernels in the spatial domain, we achieve computational efficiency without compromising performance. This approach significantly reduces GPU memory usage and computational overhead.
>
> 3. **Enhanced Translation Invariance:**
> Previous research, such as Azulay & Weiss (2019), has shown that downsampling and max-pooling can disrupt translation invariance. Other studies (e.g., [5]) have used adaptive max-pooling to preserve it. Our method achieves translation invariance theoretically and enhances it in practice through the use of 5×5 kernels, downsampling feature maps, and conventional max-pooling.
>
> 4. **Scalable Periodic Kernels in Log-Scale:**
> By assuming that the log-scale of kernels is periodic, we enable scalability by manipulating continuous scaling with a finite set of discrete representations. Although periodicity in scale may not have a direct physical interpretation, it is practically useful. This assumption eliminates the need to sample scale frequencies extensively, thus enhancing computational efficiency.
>
> **Summary:**
>
> While Zhang & Williams (2022) introduced the concept of similarity group convolution using AFMT basis in an arXiv preprint (not peer-reviewed), our work significantly advances this idea by:
> * Developing deep similarity group convolutional networks.
> * Implementing an efficient spatial domain approach with cropped Fourier series kernels.
> * Demonstrating enhanced translation invariance and scalability.
> * Validating our methods on popular benchmark datasets, showcasing practical improvements over prior work.
>
> We believe these contributions address the limitations of Zhang & Williams (2022) and represent a meaningful advancement in the field.
>
> [5] Zhang, Richard. "Making convolutional networks shift-invariant again." International conference on machine learning. PMLR, 2019.

---

> ### Author Response · Authors · 2024-11-25
>
> > 2. Does the Fourier transform approach of the AFMT basis functions scale well? Are there other advantages of working in this representation scheme? The experiments presented are on MNIST and CIFAR10/CIFAR100 datasets, but would this approach be more useful in non-image domains?
>
> To answer these questions, we would like to restate why and how our work advances this field.
>
> * **Application Scenario**: The primary application scenario of our method is computer vision, where the inputs are 2D images. Real-world objects are three-dimensional, but camera sensors capture 2D images. The motions of cameras and objects result in transformations in the captured 2D images, including translation, rotation, and uniform scaling. Additionally, these transformations are continuous, yet images are discrete samples of continuous signals.
>
> * **Efficient Handling of Continuous Transformations:**
> The shiftable, steerable, and scalable filters proposed by Freeman et al. (1991) and Simoncelli et al. (1992) are 2D filters that can be continuously transformed. This allows the resulting feature maps to be represented as linear combinations of a set of feature maps. In this way, we can handle continuous translation, rotation, and scaling using finite and discrete representations.
>
> * **Learned Filters with Theoretical Foundations:**
> However, these filters are handcrafted. Circular harmonics, which are both steerable filters and basis functions, were used in neural networks so that steerable filters could be learned from data (Worrall et al., 2017). We advance this field by introducing the Fourier series of AFMT basis functions, which are shiftable, steerable, scalable, and serve as basis functions, integrating them into deep neural networks. This enables us to handle continuous translation, rotation, and scaling using finite and discrete representations.
>
> * **Feature Matching in Structure-from-Motion:**
> One promising application of our method, as also asked by other reviewers, is feature matching. Feature matching is the task of finding corresponding points between images taken from different perspectives. It is the cornerstone of the structure-from-motion pipeline, which reconstructs a 3D model from a set of 2D images with unknown camera poses. Recent feature matching methods use conventional CNNs to train a Siamese network on pairs consisting of an image and its transformed version. The transformations used for data augmentation involve translation, rotation, and uniform scaling [6–8]. Therefore, equivariance to the similarity group (without reflection) is a desirable property in this application.
>
> > Does the Fourier transform approach of the AFMT basis functions scale well?
>
> We have seen the two works [1, 2] you mentioned that can achieve equivariance in arbitrary dimensions. However, this is not the focus of our method, nor is it the focus of the field we aim to advance. We do not know if our approach will work in 3D or higher-dimensional spaces.
>
> > Are there other advantages of working in this representation scheme?
>
> Yes. Our method use a Fourier basis, which is a complete basis, to learn kernels. It can achieve exact equivariance in theory. As our method is primarily intended for use in computer vision tasks. In these tasks, the data type is uniformly sampled 2D images. We do not need to achieve arbitrary Lie group equivariance on higher-dimensional and non-uniformly sampled data. Therefore, exact equivariance is what we desire.
>
> Another advantage is explainability. Our method employs Fourier analysis and deterministic integration, grounded in well-established theories like the Fourier Transform and Nyquist–Shannon Sampling Theorem. We interpret similarity group equivariance in terms of signal bandwidth and joint localization (see Lines 304–308), providing clear insights into the model's behavior.
>
> > The experiments presented are on MNIST and CIFAR10/CIFAR100 datasets, but would this approach be more useful in non-image domains?
>
> In our primary focus, i.e. computer vision tasks, we are dealing with 3D real-world objects and 2D images, which involves information loss due to dimensionality reduction. We expect that this line of work can inspire methods for similar scenarios—namely, dealing with low-dimensional data to accomplish tasks in higher-dimensional spaces.
>
> [6] DeTone, Daniel, Tomasz Malisiewicz, and Andrew Rabinovich. "Superpoint: Self-supervised interest point detection and description." Proceedings of the IEEE conference on computer vision and pattern recognition workshops. 2018.
>
> [7] Sarlin, Paul-Edouard, et al. "Superglue: Learning feature matching with graph neural networks." Proceedings of the IEEE/CVF conference on computer vision and pattern recognition. 2020.
>
> [8] Gleize, Pierre, Weiyao Wang, and Matt Feiszli. "Silk: Simple learned keypoints." Proceedings of the IEEE/CVF international conference on computer vision. 2023.

---

> ### Author Response · Authors · 2024-12-02
> **Modification Summary of Revised Paper for Reviewer WwDC**
>
> Thanks to the valuable reviews from all reviewers, we have updated and uploaded the manuscript. For your particular concerns, contributions relative to [1-3] and advantages of working in this representation scheme, we provide more comparisons after the summary.
>
> * **Introduction:** We added a paragraph introducing the Lie group and Lie algebra approach and discussed its limitations in achieving continuous translation equivariance. Currently, methods in both approaches focus on continuous equivariance with respect to the group $H$ of {$R^n, +$} $\rtimes H$. Specifically, these methods focus on continuous transformations $A^{-1} x$ in the $R^2$ plane (Eq. (3) of our manuscript). This is primarily because the translation group {$R^n, +$} is a non-compact group. Our approach extends the scope of continuous transformations from $A^{-1}x$ to $A^{-1} x – y$. Additionally, e refined the contribution list, highlighting our contributions relative to Zhang & Williams (2022) or [3], as well as the application scenarios, such as computer vision tasks involving 2D images.
>
> * **Related Works:** We have enhanced the discussion of the Lie group and Lie algebra approach by including more recent works. Furthermore, we explicitly compare our method’s contribution relative to these two approaches. Compared to methods in the steerable CNNs approach, we derive a Fourier basis for the similarity group and demonstrate the effectiveness of this basis by applying a deep SECNN to image classification. Compared to methods in both the steerable CNNs and Lie group Lie algebra approaches, our method achieves continuous translation equivariance because we derive the Fourier transform of the basis functions defined in the log-polar coordinate system, which allows us to construct a Fourier series for these functions in the $R^2$ plane. Lastly, we have added more works to Table 1 for a more comprehensive comparison.
>
> * **Translated, Rotated and Scaled MNIST Experiment:** We have added the baseline sim2CNN [Knigge et al., 2022] to this experiment. For the sake of brevity, we report results from only one run for each dataset due to the revision deadline. Additionally, we did not match the number of parameters of sim2CNN with the others. This decision was made because [Knigge et al., 2022] claimed state-of-the-art results on rotated MNIST, and we used this as a baseline to show that even a small amount of (non-integer) translation can significantly degrade performance.
> * **Background: Shiftable, Steerable and Scalable Filters** We have clarified that shiftable, steerable, and scalable filters are, in fact, steerable filters with respect to translation, rotation, and scaling.
>
> > contributions relative to [1-3] and advantages of working in this representation scheme
>
> In addition to including the non-compact scale group, which is not included in [1-2], our method enables us to incorporate continuous translation equivariance, which is missing in all previous equivariant literary to the best of our knowledge.
>
> [1] Implicit Convolutional Kernels for Steerable CNNs, Zhdanov et al
>
> [2]Fast, Expressive Equivariant Networks through Weight-Sharing in Position-Orientation Space, Bekkers et al
>
> [3] Similarity Equivariant Linear Transformation of Joint Orientation-Scale Space Representations, Zhang et al

---

### Official Review · Reviewer_wp2B · 2024-11-04

**Soundness:** 4
**Presentation:** 4
**Contribution:** 4
**Rating:** 8
**Confidence:** 4

**Summary:**

This work presents SECNNs, a family of neural networks equivariant to continuous translation, rotation and scale, or to discrete similarity groups. Their implementation encompasses the use of steerable, approx. shiftable and scalable basis for the convolutional kernels. In their evaluation, authors show remarkable experimental results for SECNNs.

**Important.** The authors claim to be first in achieving these equivariances, but this is not true. There is in fact a work from ICML 2022 that already provided it (in fact it can achieve equivariance to Sim(2) without the constraints (continuous vs. discrete) of this work) [1] .

**Strengths:**

- The method presented here is novel, with multiple important contributions. I believe it has tons of potential for the field, both for future research and practical applications.

- Despite having multiple theoretical contributions, the paper is very easy to follow. It is clearly written and well-structured. I enjoyed reading the paper very much!

**Weaknesses:**

- I did not observe any strong weaknesses in this paper. However, there is one important point that must be addressed / modified in the paper.

  In particular, the paper claims that it is the first method ever to achieve equivariance to Sim(2). However, there have been previous works that tackle (and provide practical implementations to this equivariance) [1]. Now, that does not demerit the contributions of this paper, as the approaches are quite different and the results are very positive, but it definitely calls for modifications in the paper positioning. I encourage the authors to do this during the rebuttal. Add proper comparisons, differences, etc. I trust that the authors will do so, and as of now, I will not let this affect my score.

- In addition, it seems that the proposed method  is not really able to achieve equivariance to the whole of the Sim(2) group –please correct me if I am wrong–. This should be stated clearly in the paper.

- Finally, as expected, the main limitation of the paper is related to memory consumption and speed. However, the authors do not provide any analysis / comparison in this aspect. It would be very helpful to provide such analyses such that the community can understand what the current cost is, e.g., in comparison to E2 nets and [1]’s method. It also would be helpful to have ablation analyses that study how the number of basis functions used affects accuracy, runtime and memory consumption.

**Questions:**

### Questions / Additional comments

- The authors do not explicitly define what seccn3D secnn4d and secnnmix are in Table 2.

- Line 317. “... allows us to precompute the basis with large enough spatial frequencies, for instance 1024x1024.” -> I am not sure I understand this. Could you please explain what you mean by this? How big will be the precomputed kernels in spatial dimensions?

- Line 355. Footnotes should go at the end of sentences after the point.

- For the Fourier Transform, it is known that its complexity grows as N log N, where N is the length of the sequence. What is the complexity / cost of going back and forth all the time from the spatial to the frequency space?

- Line 445-449. Is this inducing separability on the kernels? If so, it would be worth noting that [1] also exploits this to scale to Sim(2).

### Other comments

- The authors mention how the inherent 4D nature of the kernels leads to very large parameter consumption. I believe that using CKConvs [2] –or subsequent parameterizations., e.g., [3, 4] – to parameterize the kernels, akin to [1] could be used to mitigate this issue. Using this parameterization could provide more flexibility / parameter efficiency, and, perhaps, leads to better results.

- [5] uses CKConvs and masking strategies to learn the level of equivariances (as opposed to E2CNNs) –Line 422. I wonder whether this approach could be used to extend [5] to truly large groups.

- Under the assumption that the scaling of the method is similar to that of the Fourier transform, e.g., NlogN, then the method's complexity would not be dependent of the kernel size. If this happens to be the case, it would be interesting to look at long context. This has gained much attention in recent years [2, 3, 4, 5, 6, 7, 8].

### Conclusion

In summary, I consider this a well-rounded paper with interesting contributions and potential impact to the field. Provided that the points related to [1] are included, I believe this paper deserves a clear acceptance. I, therefore, recommend this paper be accepted.


### References

[1] Knigge, D.M., Romero, D.W. and Bekkers, E.J., 2022, June. Exploiting redundancy: Separable group convolutional networks on lie groups. In International Conference on Machine Learning (pp. 11359-11386). PMLR.

[2] Romero, D.W., Kuzina, A., Bekkers, E.J., Tomczak, J.M. and Hoogendoorn, M., 2021. Ckconv: Continuous kernel convolution for sequential data. arXiv preprint arXiv:2102.02611.

[3] Romero, D.W., Bruintjes, R.J., Tomczak, J.M., Bekkers, E.J., Hoogendoorn, M. and van Gemert, J.C., 2021. Flexconv: Continuous kernel convolutions with differentiable kernel sizes. arXiv preprint arXiv:2110.08059.

[4] Knigge, D.M., Romero, D.W., Gu, A., Gavves, E., Bekkers, E.J., Tomczak, J.M., Hoogendoorn, M. and Sonke, J.J., 2023. Modelling Long Range Dependencies in $ N $ D: From Task-Specific to a General Purpose CNN. arXiv preprint arXiv:2301.10540.

[5] Romero, David W., and Suhas Lohit. "Learning partial equivariances from data." arXiv preprint arXiv:2110.10211 (2021).

[6]  Gu, A., Goel, K. and Ré, C., 2021. Efficiently modeling long sequences with structured state spaces. arXiv preprint arXiv:2111.00396.

[7] Poli, M., Massaroli, S., Nguyen, E., Fu, D.Y., Dao, T., Baccus, S., Bengio, Y., Ermon, S. and Ré, C., 2023, July. Hyena hierarchy: Towards larger convolutional language models. In International Conference on Machine Learning (pp. 28043-28078). PMLR.

[8] Moskalev, A., Prakash, M., Liao, R. and Mansi, T., 2024. SE (3)-Hyena Operator for Scalable Equivariant Learning. arXiv preprint arXiv:2407.01049.

---

> ### Author Response · Authors · 2024-11-25
>
> Thank you for your helpful feedback; we will definitely revise the submission to better position it within the existing literature.
>
> > In particular, the paper claims that it is the first method ever to achieve equivariance to Sim(2). However, there have been previous works that tackle (and provide practical implementations to this equivariance) [1]. Now, that does not demerit the contributions of this paper, as the approaches are quite different and the results are very positive, but it definitely calls for modifications in the paper positioning. I encourage the authors to do this during the rebuttal. Add proper comparisons, differences, etc. I trust that the authors will do so, and as of now, I will not let this affect my score.
>
> We agree that [1] has achieved continuous translation, rotation, and scaling equivariance. The differences between [1] and our submission are as follows:
>
> 1. **Reflection Equivariance:** In addition to translation, rotation and scaling, our SECNNs also achieve reflection equivariance.
> 2. **Learning Convolution Kernels:** Reference [1] advances the line of work on learning group convolution kernels via MLPs (Finzi et al., 2020) to SIREN. Our work progresses from handcrafted filters (Freeman et al., 1991; Simoncelli et al., 1992) , steerable circular harmonic basis functions (Worrall et al., 2017) and steerable cnns (Weiler and Cesa., 2019) to AFMT basis functions. The two approaches have their own strengths: the former aims for arbitrary Lie groups and can deal with non-uniformly sampled data, while the latter provides explainability in terms of the Fourier transform and the Nyquist–Shannon sampling theorem. Additionally, the latter can learning kernels that construct networks that are exactly equivariant to continuous groups whereas the former only approximate these kernels using MLPs or SIREN.
> 3. **Non-Separable Kernels:** Reference [1] assumes that kernels are separable and use SIREN as kernel parameterization, increasing computational efficiency. This is an approximation that based on the separable assumption, however. The basis function defined in Eq. (13) in our submission is not separable (see the terms $(\omega_\phi - \omega_\theta, \omega_\rho - \omega_r)$). Thus, it is more general than assuming separable kernels. However, our theoretical framework is general enough to include the separable case.
>
> > In addition, it seems that the proposed method is not really able to achieve equivariance to the whole of the Sim(2) group –please correct me if I am wrong–. This should be stated clearly in the paper.
>
> This is described at lines 347–348; however, we will make it clearer in the introduction and experiment sections.
>
> > Finally, as expected, the main limitation of the paper is related to memory consumption and speed. However, the authors do not provide any analysis / comparison in this aspect. It would be very helpful to provide such analyses such that the community can understand what the current cost is, e.g., in comparison to E2 nets and [1]’s method. It also would be helpful to have ablation analyses that study how the number of basis functions used affects accuracy, runtime and memory consumption.
>
> We agree that the high computational cost is currently the main challenge in this field if we want to achieve continuous equivariance over a large group. This needs to be improved for group convolutional networks to be used as backbone architectures in various applications.
>
> Here are the computational costs we measured in the CIFAR experiments: SECNN runs on 8 GPUs, whereas other baselines run on 1 GPU.
>
> * **Speed (seconds / epoch):** WRN 23, SESN 93, E2CNN 128, SECNN 355
>
> * **GPU memory (GB):** WRN 8, SESN 8, E2CNN 11, SECNN 328
>
> > Line 355. Footnotes should go at the end of sentences after the point.
>
> Will do.
>
> >  For the Fourier Transform, it is known that its complexity grows as N log N, where N is the length of the sequence. What is the complexity / cost of going back and forth all the time from the spatial to the frequency space?
>
> The complexity of the Fourier Transform is $O(N \log N)$ primarily when the discrete sample length is $2^n$. In our case, the number of frequencies $M$ is 5, and the number of samples $N$ is 8. Thus, the complexity is $O(MN)$. There is much we can do to fine-tune these parameters; however, our primary focus was on demonstrating the theory while achieving optimal performance as much as possible.
>
> > Line 445-449. Is this inducing separability on the kernels? ...
>
> Yes, this does induce separability on the kernels. We will mention [1] in our revised manuscript.

---

> ### Author Response · Authors · 2024-11-25
>
> > The authors mention how the inherent 4D nature of the kernels leads to very large parameter consumption. I believe that using CKConvs [2] –or subsequent parameterizations., e.g., [3, 4] – to parameterize the kernels, akin to [1] could be used to mitigate this issue. Using this parameterization could provide more flexibility / parameter efficiency, and, perhaps, leads to better results.
>
> Our approach is primarily a Fourier analysis of the similarity group, providing explainability in terms of frequency and exact similarity equivariance. Moreover, our method is primarily intended for use in computer vision tasks. In these tasks, the data type is uniformly sampled 2D images. We do not need to achieve arbitrary Lie group equivariance on higher-dimensional and non-uniformly sampled data. Therefore, exact equivariance is what we desire.
>
> > The authors do not explicitly define what seccn3D secnn4d and secnnmix are in Table 2.
>
> These architectures are described in Table 4 in the Appendix. We mention this at lines 480–481, but we will update the caption of Table 2 to refer to Table 4 for clarity.
>
> > Line 317. “... allows us to precompute the basis with large enough spatial frequencies, for instance 1024x1024.” -> I am not sure I understand this. Could you please explain what you mean by this? How big will be the precomputed kernels in spatial dimensions?
>
> If we were to implement the group convolutions in Equations (14–17) solely in the frequency domain, the convolution over $\mathbb{R}^2$ would become a multiplication in the frequency domain. To do this, we would need to compute the frequency components of the images. Take a $32 \times 32$ CIFAR image as an example. If we compute the Discrete Fourier Transform (DFT) of the image—which corresponds to discrete convolution—we obtain $32 \times 32$ frequency components.
>
> We could, in theory, compute the Fourier coefficients (which correspond to continuous translation) of the image up to $1024 \times 1024$ frequencies to capture finer details. However, this is impractical due to the excessive computational cost involved in handling such a large number of frequencies.
>
> For this reason, we decided to implement the convolution over $\mathbb{R}^2$ in the spatial domain by cropping the Fourier series to $5 \times 5$ patches. This means that instead of computing a large number of frequency components, we focus on a small, manageable spatial region, which significantly reduces computational demands.
>
> Our method assumes band-limited Fourier series, meaning that frequencies outside a certain bandwidth are zero. However, the Fourier coefficients we derived in Equation (9) are not strictly band-limited; they include a decaying radial function $\bar{\rho}^{,-2 - \alpha_\rho}$. In our case, $\alpha_\rho = -1$, so the function becomes $\bar{\rho}^{,-1}$. This function approaches zero as $\bar{\rho}$ increases but never actually reaches zero (as shown in Fig. 1(c)).
>
> Despite this, we consider this behavior as effective bandlimiting because, beyond a certain point, the values become negligibly small and do not significantly contribute to the convolution. We chose the number 1024 for the spatial frequencies because we believed it was sufficiently large to capture the necessary details while remaining computationally feasible. Precomputing the $5 \times 5$ spatial patches using the inverse Fourier transform (or more precisely, the inverse Fourier series) is not resource-intensive and needs to be done only once.
>
> In summary, the precomputed kernels in the spatial domain are $5 \times 5$ in size. We focus on this small spatial region to make the computations practical while still effectively capturing the necessary frequency information for our convolution operations.
>
> >[5] uses CKConvs and masking strategies to learn the level of equivariances (as opposed to E2CNNs) –Line 422. I wonder whether this approach could be used to extend [5] to truly large groups.
>
> We are not sure if our approach can be used to extend [5]. We will attempt to interpret [5] from the perspective of our method; please correct us if we are mistaken. The partial group convolution illustrated in Fig. 1 of [5] is exactly what occurs in scale group convolution. Our method assumes periodicity on the logarithmic scale and uniformly samples this periodic log-scale to avoid partial group convolution. If we allow partial group convolution, then the masking strategy effectively multiplies a signal with a translated rectangular function. This operation is equivalent to convolving the corresponding signal in the frequency domain with a sinc function.
>
> > Under the assumption that ...
>
> We can carefully choose the number of frequencies and the number of samples such that $N = 2^n$, allowing us to transform the signal back and forth between the spatial domain and the frequency domain using the Fast Fourier Transform (FFT). This approach enables efficient computations because the FFT has a complexity of N log(N)

---

> ### Author Response · Authors · 2024-12-02
> **Modification Summary of Revised Paper for Reviewer wp2B**
>
> Thanks to the valuable reviews from all reviewers, we have updated the manuscript as follows:
>
> * **Introduction:** We added a paragraph introducing the Lie group and Lie algebra approach and discussed its limitations in achieving continuous translation equivariance. Currently, methods in both approaches focus on continuous equivariance with respect to the group $H$ of {$R^n, +$} $\rtimes H$. Specifically, these methods focus on continuous transformations $A^{-1} x$ in the $R^2$ plane (Eq. (3) of our manuscript). This is primarily because the translation group {$R^n, +$} is a non-compact group. Our approach extends the scope of continuous transformations from $A^{-1}x$ to $A^{-1} x – y$. Additionally, e refined the contribution list, highlighting our contributions relative to Zhang & Williams (2022), as well as the application scenarios, such as computer vision tasks involving 2D images.
>
> * **Related Works:** We have enhanced the discussion of the Lie group and Lie algebra approach by including more recent works. Furthermore, we explicitly compare our method’s contribution relative to these two approaches. Compared to methods in the steerable CNNs approach, we derive a Fourier basis for the similarity group and demonstrate the effectiveness of this basis by applying a deep SECNN to image classification. Compared to methods in both the steerable CNNs and Lie group Lie algebra approaches, our method achieves continuous translation equivariance because we derive the Fourier transform of the basis functions defined in the log-polar coordinate system, which allows us to construct a Fourier series for these functions in the $R^2$ plane. Lastly, we have added more works to Table 1 for a more comprehensive comparison.
>
> * **Translated, Rotated and Scaled MNIST Experiment:** We have added the baseline sim2CNN [Knigge et al., 2022] to this experiment. For the sake of brevity, we report results from only one run for each dataset due to the revision deadline. Additionally, we did not match the number of parameters of sim2CNN with the others. This decision was made because [Knigge et al., 2022] claimed state-of-the-art results on rotated MNIST, and we used this as a baseline to show that even a small amount of (non-integer) translation can significantly degrade performance.
>
> * **Background: Shiftable, Steerable and Scalable Filters** We have clarified that shiftable, steerable, and scalable filters are, in fact, steerable filters with respect to translation, rotation, and scaling.
>
> We show additional runtime of sim2CNN and SECNN-4d on MNIST with 128 batch size. sim2CNN runs on 1×A100, SECNN-4d runs on 4×A100. As sim2CNN was implemented to run on a single GPU, we assume that it achieves four times speedup on 4 GPUs.
>
> | |sim2CNN | SECNN-4d |
> |--- |--- | ---|
> |Parameters (Million) | 0.79 | 2.57|
> |Speed (seconds / epoch) | 48 (12 with 4x seedup) |  8 |
> | GPU memory (GB) | 16 | 21|
>
> > Important. The authors claim to be first in achieving these equivariances, but this is not true. There is in fact a work from ICML 2022 that already provided it (in fact it can achieve equivariance to Sim(2) without the constraints (continuous vs. discrete) of this work) [1] .
>
> The constraints (continuous vs. discrete) result from the discrete transformation, reflection. Consequently, the translation-reflection group (or Dihedral group) is a discrete group, and there is no continuous Dihedral group. However, this does not necessarily mean that the $A^{-1} x - y$ transformation on the $R^2$ plane is discrete. It is still continuous. The concept of discrete is about discretely sampling $\phi, \theta \in S^1$ in Eq. (3). The Sim(2) of [1] does not include reflection.

---

### Official Review · Reviewer_uMhw · 2024-11-07

**Soundness:** 2
**Presentation:** 2
**Contribution:** 1
**Rating:** 5
**Confidence:** 3

**Summary:**

The paper describes a group convolution architecture that is equivariant to the similarity group. It is based one expanding the convolution kernels into a (Fourier-Melin) basis that is steerable w.r.t. to the rotation, scale and reflection action. The method is evaluated on a variation of MNIST and on CIFAR10.

**Strengths:**

## Strengths
1. The paper presents what appears to be the first rotation-scale equivariant group convolution architecture utilizing this particular steerable approach. While previous work [Sosnovik et al. 2020] implemented steerable scale-equivariant G-CNNs, they employed a different definition of steerability.
2. The paper provides a thorough construction of the steerable basis, building upon the foundation established in [Zhang and Williams 2022].

**Weaknesses:**

## Areas for Improvement and Suggestions

### Historical Context and Novelty Claims
The paper's claim of being the first to implement SIM(2) equivariant networks (emphasized in the abstract's first line) would benefit from more precise positioning within the literature. [Knigge et al. 2022] previously proposed this using the regular group convolution paradigm, with arguably more efficient implementation due to its separable nature. Additionally, the relationship to [Zhang and Williams 2022] could be more clearly articulated, as they also address similarity equivariant networks.

### Technical Contributions
The extension from [Zhang and Williams 2022] appears primarily focused on the addition of reflection. It would be helpful to clarify the significance of this contribution, particularly regarding the SECNN 3D and 4D implementations shown in the tables (presumably versions without flips). It seems that in Zhang and Williams the theory is already presented, and if the only contribution is the flips (which is trivial), and even more if they are not used in the experiments, I don't really see any contribution.

### Experimental Validation
The practical advantages of the method could be more thoroughly demonstrated. While showing improved performance on transformed MNIST, the method's performance on CIFAR falls below other baselines. Given the increased computational complexity, a more detailed discussion of the method's optimal use cases would be valuable.

### Technical Clarifications Needed

1. Terminology: The distinction between shiftable, steerable, and scalable would benefit from more precise definitions, particularly regarding the specific group representations involved. That is, in my point of view these terms are collectively referred to as steerable, but then one can define steerability w.r.t. different groups and different group representations.

2. Implementation Details:
   - Line 64: Please specify the axes of the 5D space
   - Line 214: Consider providing interpretations for variables $\omega_\phi$, $\alpha_\rho$, and $\omega_\rho$
   - The nature of the "approximate" steerability requires clarification. Approximate is mentioned various time but it is never explain in what sense there is an approximation.
   - Equations 6 and 14 appear to have inconsistencies regarding position dependence in $M{g}$. Equation 6 seems that M{g} provides the frequencies, but equation 14 also shows a dependency on positions.

3. Theoretical Framework:
   - The paper appears to generate steerable feature fields through basis function convolution, placing it within the steerable group convolutions framework (cf. [Weiler and Cesa 2019])
   - The 5x5 grid implementation raises questions about effectively modeling scale variations. That is, it is reasonable to assume that sensible scale variations can be modeled on a small 5x5 grid? How are other scale equivariant methods handling this?
   - Line 330's equation setting $x_0, x_1$ equal to $\phi, \rho$ is not quite correct. It is not an equality but rather equivalence (reparametrization). I mean $x_0$ is not equal to $\phi$.
   - I do not understand the logic in line 341: " As the factor ..."

### Additional Literature
Consider incorporating discussion of:
- [Marcos et al. 2018]'s work on scale equivariance too when citing Marcos et al.'s work on rotation equivariance.
- [Knigge et al. 2022] in the introduction and Table 1
- The scale equivariant Lie group method [Bekkers 2019] in Table 1.

### Technical Details Requiring Clarification
1. Equation 14: Please clarify the distinction between $f$ and $f'$, what does the prime indicate?
2. Line 402's observation about frequency domain implementation is great and it aligns with [Cesa and Weiler 2019]'s findings. It also seems to motive more recent works like [Bekkers et al. 2024] to completely avoid Fourier based steerable methods.
3. Line 426: Please specify which axes of the 5D space constitute the "4D nature"
4. Table 2: The terms 3D, 4D, and mixed need explicit definition regarding symmetries
5. Line 498: The SECNN-Mix weight blending process requires elaboration, what is meant here?

### Experimental Evaluation
Consider including comparisons based on equal channel counts, even if this results in different parameter counts, to provide additional insight into the method's efficiency. That is, I don't believe matching parameters is necessarily a fair thing to do.

## References

Bekkers, E. J., Vadgama, S., Hesselink, R., Van der Linden, P. A., & Romero, D. W. (2024). Fast, Expressive SE(n) Equivariant Networks through Weight-Sharing in Position-Orientation Space. In The Twelfth International Conference on Learning Representations.

Knigge, D. M., Romero, D. W., & Bekkers, E. J. (2022, June). Exploiting redundancy: Separable group convolutional networks on lie groups. In International Conference on Machine Learning (pp. 11359-11386). PMLR.

Marcos, D., Kellenberger, B., Lobry, S., & Tuia, D. (2018). Scale equivariance in CNNs with vector fields. arXiv preprint arXiv:1807.11783.

Sosnovik, I., Szmaja, M., & Smeulders, A. (2020). Scale-Equivariant Steerable Networks. In International Conference on Learning Representations.

Weiler, M., & Cesa, G. (2019). General e(2)-equivariant steerable CNNs. Advances in Neural Information Processing Systems, 32.

Zhang, X., & Williams, L. R. (2022). Similarity equivariant linear transformation of joint orientation-scale space representations. arXiv preprint arXiv:2203.06786.

**Questions:**

see above.

---

> ### Author Response · Authors · 2024-11-26
>
> > Additionally, the relationship to [Zhang and Williams 2022] could be more clearly articulated, as they also address similarity equivariant networks.
>
> We apologize for any confusion caused by the way we presented our work.  Zhang et al. (2022) is an arXin preprint but not a peer-reviewed paper.
>
> **Limitations of Zhang & Williams (2022):**
> 1. **Limited Experimental Scope and Shallow Implementation:**
> Zhang & Williams (2022) conducted experiments involving only a single convolution with a precomputed kernel. This shallow implementation does not fully explore the potential of similarity group convolutions in deep neural networks.
>
> 2. **Inefficient Frequency Domain Implementation:**
> Their implementation of group convolution exclusively in the frequency domain proves to be computationally inefficient and limits performance. For example, processing a $32\times 32$ CIFAR image requires handling all $32\times 32$ frequency components (or more), resulting in significantly higher GPU memory consumption compared to our proposed spatial domain implementation, which uses cropped $5\times 5$ Fourier series.
>
> 3. **Lack of Scalability Due to Non-Periodic Kernels:**
> The kernels are not assumed to be periodic with respect to the logarithmic scale, which hampers scalability. Without periodicity, continuous scaling requires extensive sampling of scale frequencies (real numbers), making it impractical for efficient computation.
>
> **Our Contributions and Improvements:**
>
> 1. **Deep Similarity Group Convolution Networks:**
> We constructed deep neural networks utilizing similarity group convolutions and tested them on popular benchmark datasets. This demonstrates the practical applicability and effectiveness of our approach in real-world scenarios.
>
> 2. **Efficient Spatial Domain Implementation:**
> While implementing group convolutions in the frequency domain is theoretically elegant, we observed that it is computationally inefficient and yields suboptimal performance (as discussed in Lines 396–405 of our submission). By using cropped $5\times 5$ Fourier series as kernels in the spatial domain, we achieve computational efficiency without compromising performance. This approach significantly reduces GPU memory usage and computational overhead.
>
> 3. **Enhanced Translation Invariance:**
> Previous research, such as Azulay & Weiss (2019), has shown that downsampling and max-pooling can disrupt translation invariance. Other studies (e.g., [1]) have used adaptive max-pooling to preserve it. Our method achieves translation invariance theoretically and enhances it in practice through the use of 5×5 kernels, downsampling feature maps, and conventional max-pooling.
>
> 4. **Scalable Periodic Kernels in Log-Scale:**
> By assuming that the log-scale of kernels is periodic, we enable scalability by manipulating continuous scaling with a finite set of discrete representations. Although periodicity in scale may not have a direct physical interpretation, it is practically useful. This assumption eliminates the need to sample scale frequencies extensively, thus enhancing computational efficiency.
>
> **Summary:**
>
> While Zhang & Williams (2022) introduced the concept of similarity group convolution using AFMT basis in an arXiv preprint (not peer-reviewed), our work significantly advances this idea by:
> * Developing deep similarity group convolutional networks.
> * Implementing an efficient spatial domain approach with cropped Fourier series kernels.
> * Demonstrating enhanced translation invariance and scalability.
> * Validating our methods on popular benchmark datasets, showcasing practical improvements over prior work.
>
> We believe these contributions address the limitations of Zhang & Williams (2022) and represent a meaningful advancement in the field.
>
> [1] Zhang, Richard. "Making convolutional networks shift-invariant again." International conference on machine learning. PMLR, 2019.

---

> > ### Comment · Reviewer_uMhw · 2024-11-26
> >
> > Thank you for these clarifications. This is very helpful. Also considering the responses below. Would it be possible to upload an updated paper with some of the clarifications and details, perhaps changes color coded. I think the rebutal is helpful, but I'd like to see how they result in improvements in the paper. I currently raised my score to 5 as I think one of the main limitations (limited contribution relative to Zhang and Williams) is somewhat alleviated.

---

> ### Author Response · Authors · 2024-11-26
>
> Thank you for your insightful feedback; we will revise the submission to better position it within the existing literature.
>
> > [Knigge et al. 2022] previously proposed this using the regular group convolution paradigm, with arguably more efficient implementation due to its separable nature.
>
> We agree that [Knigge et al. 2022] has achieved continuous translation, rotation, and scaling equivariance.The differences between [Knigge et al. 2022] and our submission are as follows:
>
> 1. **Reflection Equivariance:** In addition to translation, rotation and scaling, our SECNNs also achieve reflection equivariance.
>
> 2. **Learning Convolution Kernels:**  [Knigge et al. 2022] advances the line of work on learning group convolution kernels via MLPs (Finzi et al., 2020) to SIREN. Our work progresses from handcrafted filters (Freeman et al., 1991; Simoncelli et al., 1992) , steerable circular harmonic basis functions (Worrall et al., 2017) and steerable cnns (Weiler and Cesa., 2019) to AFMT basis functions. The two approaches have their own strengths: the former aims for arbitrary Lie groups and can deal with non-uniformly sampled data, while the latter provides explainability in terms of the Fourier transform and the Nyquist–Shannon sampling theorem. Additionally, the latter can learning kernels that construct networks that are exactly equivariant to continuous groups whereas the former only approximate these kernels using MLPs or SIREN.
>
> 3. **Non-Separable Kernels:** [Knigge et al. 2022] assumes that kernels are separable and use SIREN as kernel parameterization, increasing computational efficiency. However, this approach involves two types of approximation: 1) separable assumption and 2) kernel parameterization. The basis defined in Eq. (13) in our submission is not separable (see the terms $(\omega_\phi - \omega_\theta, \omega_\rho - \omega_r)$) and it is a complete basis. Thus, it is more general. Additionally, our theoretical framework is also general enough to include the separable case (see line 445).
>
> > Given the increased computational complexity, a more detailed discussion of the method's optimal use cases would be valuable.
>
> * **Application:** Our method is primarily intended for use in computer vision tasks. In these tasks, the data type is uniformly sampled 2D images. We do not need to achieve arbitrary Lie group equivariance on higher-dimensional and non-uniformly sampled data. Therefore, exact equivariance is what we desire. One promising application of our method, as also asked by other reviewers, is feature matching. Feature matching is the task of finding corresponding points between images taken from different perspectives. It is the cornerstone of the structure-from-motion pipeline, which reconstructs a 3D model from a set of 2D images with unknown camera poses. Recent feature matching methods use conventional CNNs to train a Siamese network on pairs consisting of an image and its transformed version. The transformations used for data augmentation involve translation, rotation, and uniform scaling [2-4]. Therefore, equivariance to the similarity group (without reflection) is a desirable property in this application.
>
> **Complexity Analysis:**
>
> * **MNIST Experiment:** We used a seven-layer SECNNs on $28\times 28$ MNIST images and the batch size is 128.
> * **Feature Matching:** [2-4] train a eight-layer CNN on $256\times 256$ COCO images and the batch size is 1.
>
> Since $(256 / 28)^2 \approx 84 < 128$, SECNNs can replace the eight-layer CNN in the feature matching task.
>
>
> [2] DeTone, Daniel, Tomasz Malisiewicz, and Andrew Rabinovich. "Superpoint: Self-supervised interest point detection and description." Proceedings of the IEEE conference on computer vision and pattern recognition workshops. 2018.
>
> [3] Sarlin, Paul-Edouard, et al. "Superglue: Learning feature matching with graph neural networks." Proceedings of the IEEE/CVF conference on computer vision and pattern recognition. 2020.
>
> [4] Gleize, Pierre, Weiyao Wang, and Matt Feiszli. "Silk: Simple learned keypoints." Proceedings of the IEEE/CVF international conference on computer vision. 2023.

---

> > ### Comment · Reviewer_uMhw · 2024-11-26
> >
> > Thank you for the clarification the distinction. I appreciate the viewpoint that the presented method does not separate along sub-group axes. Arguably, the paper by Knigge et al. could also be implemented without separating the kernels, but at the cost of computational complexity. I appreciate the effort made in placing the current work in context to literature.

---

> ### Author Response · Authors · 2024-11-26
>
> > 1. Terminology: The distinction between shiftable, steerable, and scalable would benefit from more precise definitions, particularly regarding the specific group representations involved. That is, in my point of view these terms are collectively referred to as steerable, but then one can define steerability w.r.t. different groups and different group representations.
>
> The concept of steerable filter was first proposed in Freeman et al. (1991).  As discussed at line 190-204, Simoncelli and Freeman (Simoncelli et al. (1992)) generalized this concept to shiftable and scalable filters (see line 190-204). After the group equivariant CNN was introduced in Cohen & Welling (2016), the same authors combined the idea of steerable filter and group equivariance CNN, proposing steerable CNNs in 2017 [5]. We followed the definitions proposed in 90s to distinguish continuous translation, rotation and scaling. We discuss this in the background section at lines 190-240.
> We appreciate your review about this, as it helps us figure out that [Sosnovik et al. 2020] uses the scale-steerable filters only, which is scalable in our terminology, instead of steerable filters (for orientation). We will avoid this potential confusing terminology by explicitly compare these two terminologies in the revised submission.
>
> [5] Cohen, Taco S., and Max Welling. "Steerable CNNs." International Conference on Learning Representations. 2017.
>
> > Line 64: Please specify the axes of the 5D space
>
> Line 214: Consider providing interpretations for variables ωϕ, αρ, and ωρ
>
> Will do.
>
> > The nature of the "approximate" steerability requires clarification. Approximate is mentioned various time but it is never explain in what sense there is an approximation.
>
> We’ve discuss why the filters are approximate shiftable and scalable at lines 304-308. We will add a subsection title to make this discussion clearer.
>
> > The paper appears to generate steerable feature fields through basis function convolution, placing it within the steerable group convolutions framework (cf. [Weiler and Cesa 2019])
>
> We’ve positioned our work in the steerable filter approach in the related works section. Due to the inconsistent terminology issue between the review and us, we will make everything clear.
>
> > The 5x5 grid implementation raises questions about effectively modeling scale variations. That is, it is reasonable to assume that sensible scale variations can be modeled on a small 5x5 grid? How are other scale equivariant methods handling this?
>
> The SESN [Sosnovik et al. 2020], which achieves scale equivariance, uses $7 \times 7$ patches. It may seem counter-intuitive that small patches, such as $5 \times 5$ or $7 \times 7$, can handle scale changes. However, our method’s theoretical framework provides an explanation from the perspective of signal processing, which we consider a strength.
>
> As shown in lines 303–304, the AFMT basis (Eq. 8) and its Fourier transform share an identical pattern. Cropping the Fourier series to $5 \times 5$ patches is equivalent to multiplying the Fourier series with a rectangular function. In the frequency domain, this operation corresponds to convolving the Fourier transform of Eq. 8 with a sinc function. This convolution results in blurred Fourier coefficients (e.g., imagine convolving the Fourier coefficients shown in the top row of Fig. 1(c) with a sinc function, which makes them blurred).
>
> Nevertheless, these $5 \times 5$ patches are still able to capture a much larger frequency range—in our case, $[-512, 511]$. It is important to recall that dilation in the spatial domain corresponds to contraction in the frequency domain, and contraction in the spatial domain corresponds to dilation in the frequency domain. Thus, even small spatial patches can effectively handle significant scale changes due to this relationship between spatial and frequency domains.
>
> > Line 330's equation setting x0,x1 equal to ϕ,ρ is not quite correct. It is not an equality but rather equivalence (reparametrization). I mean x0 is not equal to ϕ.
>  I do not understand the logic in line 341: " As the factor ..."
>
> At line 214, we define the origin of $(\phi, \rho)$ to be $(x_0, x_1) = (0, 0)$. As it is shown in Fig. 1, the origin locates at the center of those patches. In other words, translations of the basis functions (Eq. 8), which are defined in the polar coordinate system, equal to translations of the actual patches that are uniformly sampled in the Cartesian coordinate system, which is shown in Fig. 1. Once this is clear, the logic in line 341 is clear. We will improve the visualization of the basis function in Fig. 1 to make the change of coordinate system clear.
>
> > Consider incorporating discussion of: [Marcos et al. 2018]'s work on scale equivariance too when citing Marcos et al.'s work on rotation equivariance. [Knigge et al. 2022] in the introduction and Table 1, The scale equivariant Lie group method [Bekkers 2019] in Table 1.
>
> Will do

---

> ### Author Response · Authors · 2024-11-26
>
> > 1. Equation 14: Please clarify the distinction between f and f′, what does the prime indicate?
>
> We are sorry that these are typo. We use f’ to denote 2D functions and f to denote 4(5)D functions. Thus, these functions should be f’ in Eq. 14.
>
> > 2. Line 402's observation about frequency domain implementation is great and it aligns with [Cesa and Weiler 2019]'s findings. It also seems to motive more recent works like [Bekkers et al. 2024] to completely avoid Fourier based steerable methods.
>
> We don't agree that the observation motivates [Bekkers et al. 2024]. This work proposes the idea of an invariant attribute $a_{ij}$, which is associated with two points $(x_i, x_j)$. This idea works when group transformations are distance-preserving. Specifically, their focus group is $\text{SE}(n)$. When it comes to scaling, which is not a distance-preserving transformation, the invariant attributes given in the paper are not valid. For example, they use Euclidean distance as the invariant attribute for translation; however, the Euclidean distance is not invariant under scaling.
>
> In our approach, the scale group is locally compact, meaning that a Haar measure is unique up to a constant multiplier. It follows that we can deal with the scale factor by performing the integral $\int \frac{dr}{r}$ (see lines 173–176). Thus, the method of Bekkers et al. 2024 would not seem superior to the Fourier approach.
>
> > 3. Line 426: Please specify which axes of the 5D space constitute the "4D nature"
>
> We implement two group equivariant networks: continuous translation, rotation and scaling group, which uses 4D weights in $R^2\times S^1 \times R^{+}$ and  continuous translation, scaling and discrete Dihedral group (orientation-reflection), which use 5D weights in $R^2\times S^1 \times R^{+} \times {\pm 1}$
>
> > 4. Table 2: The terms 3D, 4D, and mixed need explicit definition regarding symmetries
> 5. Line 498: The SECNN-Mix weight blending process requires elaboration, what is meant here?
>
> We list the corresponding architecture in Table 4 in Appdenix. We will make the symmetries clear in the caption of Table 3, and refer the Table 4 when we discuss SECNN-Mix.

---

> > ### Comment · Reviewer_uMhw · 2024-11-26
> >
> > Regarding the above item (2.), it was not my intention to induce a discussion on superiority of methods relative to each other. Each method has their own merits. The point I wanted to make is that when it comes to the comparison of Fourier-based (steerable) methods and regular group conv methods, it seems that the latter is practically often preferred since the point-wise activation functions that are allowed in that framework are more powerful than norm-based activation functions that have to be used with Fourier-based methods. It seems that in you work also shows that activations in normal space (non-frequency/irrep domain) is benificial, and I think this is a nice contribution.
> >
> > I dragged in the reference Bekkers et al. as it fully circumvents the need for Fourier-based (tensor field type) methods for continuous rotation equivariance and as such by-passes the need for Fourier transforms all together. Their appendix elaborates on this perspective.

---

> > > ### Author Response · Authors · 2024-11-26
> > >
> > > > Regarding the above item (2.), it was not my intention to induce a discussion on superiority of methods relative to each other. Each method has their own merits. The point I wanted to make is that when it comes to the comparison of Fourier-based (steerable) methods and regular group conv methods, it seems that the latter is practically often preferred since the point-wise activation functions that are allowed in that framework are more powerful than norm-based activation functions that have to be used with Fourier-based methods. It seems that in you work also shows that activations in normal space (non-frequency/irrep domain) is benificial, and I think this is a nice contribution.
> > >
> > > >I dragged in the reference Bekkers et al. as it fully circumvents the need for Fourier-based (tensor field type) methods for continuous rotation equivariance and as such by-passes the need for Fourier transforms all together. Their appendix elaborates on this perspective.
> > >
> > > I see your point now. Their argument is that transforming back and forth between the spatial and frequency domain may cause errors. The Fourier approach should not have such a problem given sufficient computational resources in theory. Yet, in practice, the errors are inevitable, especially when we have to deal with non-linear operations, which is what we have observed. However, we would like to point out that the Fourier approach enables us to achieve continuous translation equivariance.
> > >
> > > Previous works that are based on either steerable filter or Lie algebra parameterize kernels regarding groups, such as $H = O(N)$ and $SO(N)$. The semidirect product between translation group and $H$ , $(R^n, +) \rtimes H$, allows these methods to apply the kernels to every position via so called point convolution. However, it is not guaranteed that this achieves continuous translation equivariance. When data is uniformly sampled on the Cartesian coordinate system, this equals to discrete convolution on the $R^2$ plane.
> > >
> > > We forgot to reply the following review.
> > >
> > > > Consider including comparisons based on equal channel counts, even if this results in different parameter counts, to provide additional insight into the method's efficiency. That is, I don't believe matching parameters is necessarily a fair thing to do.
> > >
> > > Just like LeCun et al. (1998) integrated convolution into neural networks to reduce parameters needed, integrating other symmetries also reduce parameters or increase parameters efficiency. Why should translation be more special than rotation and scaling? I believe it should not be, even though images are uniformly sampled on a rectangular grid. This is why previous works, such as SESN, E2CNN, etc., maintain a similar number of parameters in their experiments. We followed this experiment design.

---

> ### Author Response · Authors · 2024-12-02
> **Modification Summary for Revised Paper for Reviewer uMhw**
>
> Thanks to the valuable reviews from all reviewers, we have updated and uploaded the manuscript.
>
> * **Introduction:** We've added a paragraph introducing the Lie group and Lie algebra approach and discussed its limitations in achieving continuous translation equivariance. Currently, methods in both approaches focus on continuous equivariance with respect to the group $H$ of {$R^n, +$} $\rtimes H$. Specifically, these methods focus on continuous transformations $A^{-1} x$ in the $R^2$ plane (Eq. (3) of our manuscript). This is primarily because the translation group {$R^n, +$} is a non-compact group. Our approach extends the scope of continuous transformations from $A^{-1}x$ to $A^{-1} x – y$. Additionally, e refined the contribution list, highlighting our contributions relative to Zhang & Williams (2022), as well as the application scenarios, such as computer vision tasks involving 2D images.
> * **Related Works:** We have enhanced the discussion of the Lie group and Lie algebra approach by including more recent works. Furthermore, we explicitly compare our method’s contribution relative to these two approaches. Compared to methods in the steerable CNNs approach, we derive a Fourier basis for the similarity group and demonstrate the effectiveness of this basis by applying a deep SECNN to image classification. Compared to methods in both the steerable CNNs and Lie group Lie algebra approaches, our method achieves continuous translation equivariance because we derive the Fourier transform of the basis functions defined in the log-polar coordinate system, which allows us to construct a Fourier series for these functions in the $R^2$ plane. Lastly, we have added more works to Table 1 for a more comprehensive comparison.
> * **Translated, Rotated and Scaled MNIST Experiment:** We have added the baseline sim2CNN [Knigge et al., 2022] to this experiment. For the sake of brevity, we report results from only one run for each dataset due to the revision deadline. Additionally, we did not match the number of parameters of sim2CNN with the others. This decision was made because [Knigge et al., 2022] claimed state-of-the-art results on rotated MNIST, and we used this as a baseline to show that even a small amount of (non-integer) translation can significantly degrade performance.
> * **Background: Shiftable, Steerable and Scalable Filters** We have clarified that shiftable, steerable, and scalable filters are, in fact, steerable filters with respect to translation, rotation, and scaling.
>
> > [Knigge et al. 2022] previously proposed this using the regular group convolution paradigm, with arguably more efficient implementation due to its separable nature.
>
> We show the runtime of sim2CNN and SECNN-4d on MNIST with 128 batch size. sim2CNN runs on 1×A100, SECNN-4d runs on 4×A100. As sim2CNN was implemented to run on a single GPU, we assume that it achieves four times speedup on 4 GPUs. The table below shows that SECNN-4d is a more efficient implementation, a consequence of implementing rotation and scaling group convolution in the frequency domain. And the GPU memory assumption is caused by the explicit sliding window function as it's discussed in the  conclusion section.
>
>
> | |sim2CNN | SECNN-4d |
> |--- |--- | ---|
> |Parameters (Million) | 0.79 | 2.57|
> |Speed (seconds / epoch) | 48 (12 with 4x seedup) |  8 |
> | GPU memory (GB) | 16 | 21|

---

### Meta-Review · Area_Chair_4f37 · 2024-12-19

**Metareview:**

In order to achieve continuous translation, rotation and scale equivariance, this paper proposes similarity group equivariant convolutional networks. Experiments on several datasets validate its effectiveness. After the response, it receives three borderline reject and one accept. The advantages, including the thorough construction of the steerable basis, the interesting idea of rotation-scale equivariant group convolution architecture, and the analysis of efficiency and ablations, are recognized by the reviewers. However, they are also concerned about the incremental contribution over [Zhang & Williams (2022)], unsatisfying presentation, insufficient experiments, etc. Though the response well addresses part of the concerns of Reviewers uMhw and WwDC, they still think the revised manuscript is marginally below the acceptance threshold. After carefully reading the paper and discussion, I also think the current manuscript does not meet the requirements of this top conference. I suggest the authors carefully revise the paper and submit it to another relevant venue.

**Additional Comments On Reviewer Discussion:**

The authors present detailed responses to the raised concerns. Some of the weaknesses are well addressed. Reviewers uMhw and WwDC raise their ratings. However, their final score is still marginally below the acceptance threshold. I also think the current manuscript does not meet the requirements of this top conference.

---

### Decision · Program_Chairs · 2025-01-22

Reject